# Stable Pom1 clusters form a glucose-modulated concentration gradient that regulates mitotic entry

Corey A H Allard, Hannah E Opalko, James B Moseley*

Department of Biochemistry and Cell Biology, The Geisel School of Medicine at Dartmouth, Hanover, United States

**Abstract** Control of cell size requires molecular size sensors that are coupled to the cell cycle. Rod-shaped fission yeast cells divide at a threshold size partly due to Cdr2 kinase, which forms nodes at the medial cell cortex where it inhibits the Cdk1-inhibitor Wee1. Pom1 kinase phosphorylates and inhibits Cdr2, and forms cortical concentration gradients from cell poles. Pom1 inhibits Cdr2 signaling to Wee1 specifically in small cells, but the time and place of their regulatory interactions were unclear. We show that Pom1 forms stable oligomeric clusters that dynamically sample the cell cortex. Binding frequency is patterned into a concentration gradient by the polarity landmarks Tea1 and Tea4. Pom1 clusters colocalize with Cdr2 nodes, forming a glucose-modulated inhibitory threshold against node activation. Our work reveals how Pom1-Cdr2-Wee1 operates in multiprotein clusters at the cortex to promote mitotic entry at a cell size that can be modified by nutrient availability.
DOI: https://doi.org/10.7554/eLife.46003.001

## Introduction

Many cell types display a remarkable ability to maintain a constant size during rapid cycles of growth and division (*Fantes and Nurse, 1977*; *Ginzberg et al., 2015*; *Jorgensen and Tyers, 2004*; *Fantes, 1977*; *Dolznig et al., 2004*; *Lloyd, 2013*). Such cell size control is a system-level property that emerges from the integration of multiple size-dependent signal transduction pathways. Each signaling pathway is comprised of tunable biochemical parameters, including gene-expression and post-translational modifications such as protein phosphorylation (*Alberghina et al., 2009*). One major challenge in cell size research is to understand the biochemical mechanisms of signal transduction in each pathway, and what makes them size-dependent. These control systems can generate size homogeneity for a given cell type, but cell size is also an adaptable property. For example, nutritional cues and other environmental factors can alter cell size (*Fantes and Nurse, 1977*; *Kelkar and Martin, 2015*; *Shiozaki, 2009*; *Yanagida et al., 2011*; *Young and Fantes, 1987*). Thus, a second major challenge in cell size research is to understand how size-dependent signaling pathways respond to changes in cell metabolism and stress.

In eukaryotic cells, these signaling pathways lead to regulated activation of the conserved cyclin-dependent kinase Cdk1 (*Harashima et al., 2013*). Activated Cdk1 triggers mitotic entry and the cascade of events that lead to cell division (*Gould and Nurse, 1989*; *Simanis and Nurse, 1986*). The fission yeast *Schizosaccharomyces pombe* is an excellent model system to study size-dependent signaling pathways that regulate Cdk1. Genetic screens performed in past decades have identified many conserved factors that regulate Cdk1, but how these factors form size-dependent signaling pathways remains less clear. Fission yeast cells have a simple geometry that facilitates cell size studies. These cylindrical cells maintain a constant cell width, and grow by linear extension during interphase (*Fantes and Nurse, 1977*; *Moreno et al., 1989*). A network of cell polarity proteins

*For correspondence:
james.b.moseley@dartmouth.edu

**Competing interests:** The authors declare that no competing interests exist.

positioned at cell tips restricts growth to these sites and maintains proper cell morphology (*Chang and Martin, 2009*). As a result, cell length doubles in one cell cycle, and many aspects of cell geometry scale with this increase in cell length (*Gu and Oliferenko, 2019*; *Neumann and Nurse, 2007*). Recent studies used cell shape mutants to show that fission yeast cells primarily measure surface area, not length or volume, for cell size control (*Facchetti et al., 2019*; *Pan et al., 2014*). A critical next step is to understand how signaling pathways that regulate Cdk1 might operate at the cell surface in a size-dependent manner.

Cdk1 activity is established by the opposing activities of the inhibitory protein kinase Wee1, and the counteracting phosphatase Cdc25 (*Gautier et al., 1991*; *Gould and Nurse, 1989*; *Kumagai and Dunphy, 1991*; *Russell and Nurse, 1986*; *Russell and Nurse, 1987*; *Strausfeld et al., 1991*). In fission yeast, mutations in Wee1, Cdc25, and their upstream regulators lead to changes in cell size. Separate mechanisms link cell size with regulation of Wee1 versus Cdc25. The cellular concentration of Cdc25 increases as cells grow during interphase (*Keifenheim et al., 2017*; *Moreno et al., 1990*). In contrast, the concentration of Wee1 remains constant during interphase, but it is progressively phosphorylated by the conserved inhibitory kinases Cdr1 and Cdr2 (*Aligue et al., 1997*; *Breeding et al., 1998*; *Kanoh and Russell, 1998*; *Keifenheim et al., 2017*; *Lucena et al., 2017*; *Opalko and Moseley, 2017*; *Russell and Nurse, 1987*; *Wu and Russell, 1993*; *Parker et al., 1993*; *Coleman et al., 1993*; *Young and Fantes, 1987*). *cdr2* mutants fail to divide at a constant surface area, and instead divide according to cell volume or length (*Facchetti et al., 2019*). This change suggests that Cdr2-Cdr1-Wee1 signaling underlies the primary size-sensing pathway that measures cell surface area, while additional pathways related to volume and length are engaged in its absence. The localization of Cdr2, Cdr1, and Wee1 support this model (*Figure 1A*): Cdr2 forms punctate oligomeric structures called nodes that stably bind to the medial cell cortex, and recruits Cdr1 to these sites (*Akamatsu et al., 2014*; *Akamatsu et al., 2017*; *Guzmán-Vendrell et al., 2015*; *Martin and Berthelot-Grosjean, 2009*; *Morrell et al., 2004*; *Moseley et al., 2009*). Wee1 localizes primarily in the nucleus and spindle-pole body, where it encounters Cdk1 to prevent mitotic entry (*Masuda et al., 2011*; *Moseley et al., 2009*; *Wu et al., 1996*). In addition, Wee1 transiently visits cortical Cdr1/2 nodes in bursts that lead to inhibition of Wee1 kinase activity (*Allard et al., 2018*). The frequency and duration of these Wee1 bursts at Cdr1/2 nodes increase approximately twenty-fold as cells double in size (*Allard et al., 2018*), leading to size-dependent inhibition of Wee1 at the cell surface.

This size-dependent change in Wee1 bursting dynamics is encoded into the Cdr1-Cdr2-Wee1 pathway at two proposed levels. First, increased Wee1 bursting depends upon the doubling in number of Cdr1/2 nodes during one cell cycle, but this 2-fold increase is smaller than the 20-fold increase in Wee1 bursting (*Allard et al., 2018*; *Deng and Moseley, 2013*; *Pan et al., 2014*). Second, activation of Cdr2 increases as cells increase in size (*Deng et al., 2014*). Cdr2 kinase activity is required for Wee1 localization to nodes, and is controlled by the upstream kinase Pom1 (*Allard et al., 2018*; *Moseley et al., 2009*). Pom1 directly phosphorylates and inhibits the activation of Cdr2 (*Bhatia et al., 2014*; *Deng et al., 2014*; *Kettenbach et al., 2015*; *Martin and Berthelot-Grosjean, 2009*; *Moseley et al., 2009*). Pom1 also phosphorylates a separate set of sites on Cdr2 to disrupt oligomerization and membrane binding (*Bhatia et al., 2014*; *Rincon et al., 2014*). Pom1 regulation of Cdr2 increases the frequency and duration of Wee1 bursts at Cdr1/2 nodes, but only in small cells (*Allard et al., 2018*). Consistent with this defect, *pom1Δ* cells divide at a small size due to dysregulation of the Cdr2-Cdr1-Wee1 pathway (*Bähler and Pringle, 1998*; *Bhatia et al., 2014*; *Martin and Berthelot-Grosjean, 2009*; *Moseley et al., 2009*; *Wood and Nurse, 2013*). Taken together, these results suggest a size-dependent interaction between Pom1 and its substrate Cdr2. However, the location and timing of Pom1 interactions with Cdr2 have remained poorly defined.

Pom1 localizes in a cortical spatial gradient that is enriched at cell tips, with a lower concentration at the medial cell cortex, where its substrate Cdr2 forms nodes (*Bähler and Pringle, 1998*; *Bhatia et al., 2014*; *Martin and Berthelot-Grosjean, 2009*; *Moseley et al., 2009*; *Bähler and Nurse, 2001*). Thus, the majority of Pom1 and Cdr2 molecules in a cell are spatially separated (*Figure 1A*), raising the question of when and where they interact. Several lines of evidence suggest that the spatial distributions of Pom1 and Cdr2 are critical for the size-dependent signaling properties of this pathway, and suggest the lateral cell cortex as the key interface. For example, ectopic targeting of Pom1 to the medial cell cortex inhibits Cdr2 node formation and Cdr2-dependent cell size signaling (*Martin and Berthelot-Grosjean, 2009*). Additionally, the Pom1 gradient is disrupted as

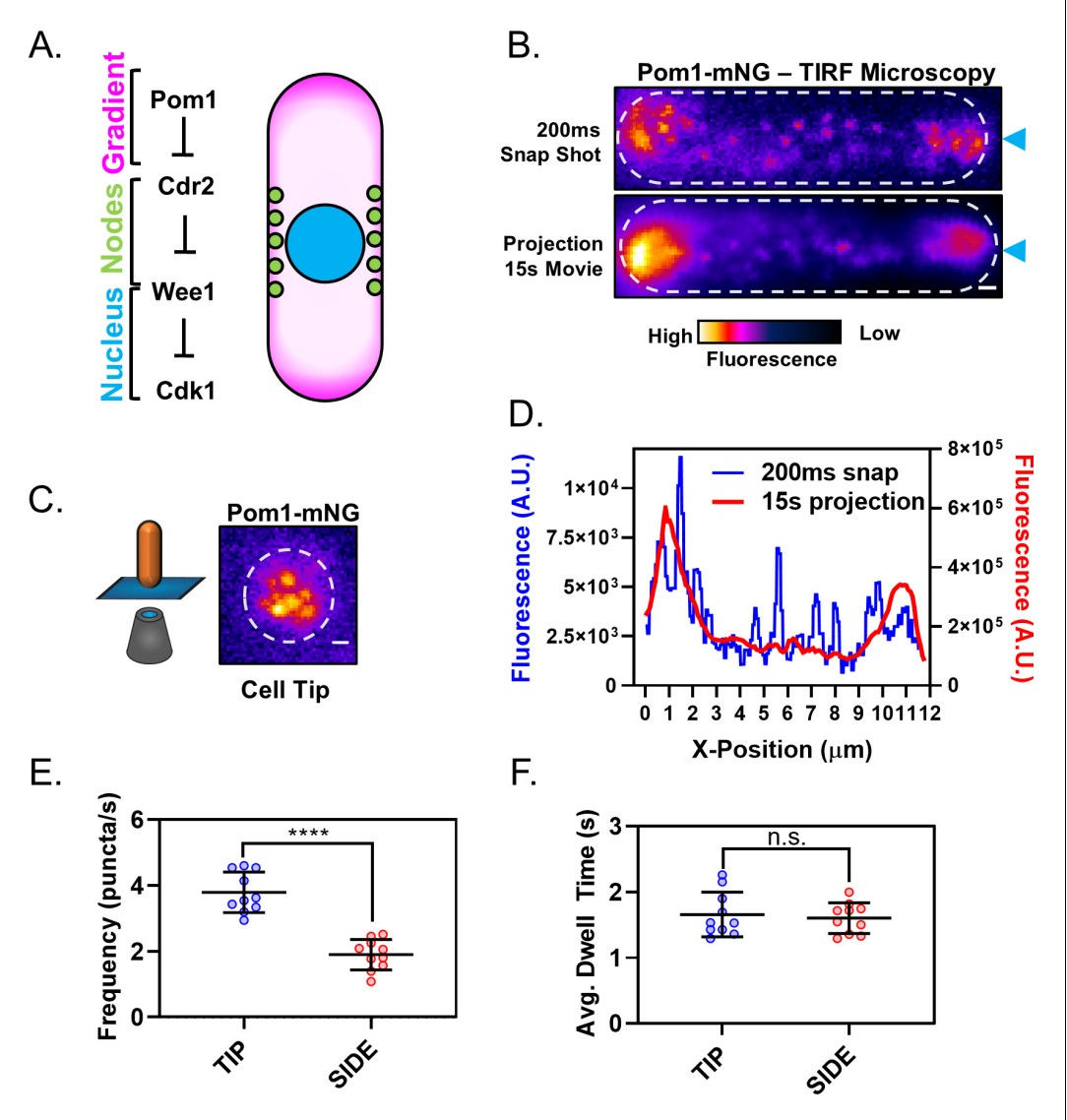

**Figure 1.** The Pom1 gradient is formed by time-averaging of clusters that transiently bind the cortex. (**A**) Schematic of the Cdr2-Pom1-Wee1 signal transduction pathway and its coarse cellular localization. (**B**) Individual frame (top panel) and sum projection (bottom panel) of a high-speed TIRF microscopy movie. Movie was continuous 15 s time-lapse acquisition of 200 ms exposures. Scale bar 1 μm. Blue arrows mark position of line scans performed for data in panel (**D**). (**C**) Pom1-mNG also forms clusters at the cell tip. Image is a sum projection of three consecutive 200 ms exposures from continuous time-lapse TIRF movie of the cell tip as depicted in the cartoon diagram. Scale bar 1 μm. (**D**) Line scans of fluorescence intensity along the long axis of the snap shot (blue line, left Y axis) and projection images (red line, right Y axis) in panel (**B**). Note that time-averaging of Pom1 clusters smoothens the concentration gradient. (**E**) Comparison of Pom1 cluster binding frequency at the cell tip or side (****p=<0.0001, n = 10 cells, 42–170 traces/cell). (**F**) Comparison of cortical dwell time of individual Pom1-mNG clusters (n.s., p=0.6747, *n* = 10 cells, 42–170 traces/cell) at the cell tip or side. For (**E–F**), each data point represents a single cell mean, and line and error bars represent mean and standard deviation of all cells. Statistical significance was tested using a Student's T-test.

DOI: https://doi.org/10.7554/eLife.46003.002

The following figure supplements are available for figure 1:

**Figure supplement 1.** Analysis of the Pom1 gradient by confocal microscopy.

DOI: https://doi.org/10.7554/eLife.46003.003

**Figure supplement 2.** Analysis of Pom1 cluster diffusion.

DOI: https://doi.org/10.7554/eLife.46003.004

part of a cellular response to glucose deprivation, which leads to increased Pom1 concentration at the lateral cortex where it delays mitotic entry (*Kelkar and Martin, 2015*). This result also demonstrates that Pom1-Cdr2 signaling responds to environmental input to coordinate cell size with nutrient availability. In this system, Pom1 functions analogously to an intracellular morphogen, acting as a concentration-dependent and localization-controlled inhibitor of Cdr2 nodes. However, levels of Pom1 at the medial cell cortex in wild-type cells growing under steady state conditions are low, constant, and do not vary with cell size (*Bhatia et al., 2014*; *Pan et al., 2014*). How Pom1 provides size-dependent input to Cdr2 is unclear and requires analysis of their molecular dynamics at the lateral cell cortex.

Past studies have led to a model for Pom1 gradient formation driven by the binding and diffusion of individual Pom1 molecules in the plasma membrane (*Hachet et al., 2011*; *Hersch et al., 2015*; *Saunders et al., 2012*). Intriguingly, dynamic clusters of Pom1 have also been observed and are proposed to form by unstable oligomerization of molecules diffusing on the membrane (*Hachet et al., 2011*; *Saunders et al., 2012*). These clusters are thought to form and decay rapidly, temporarily slowing Pom1 diffusion rates and reducing noise in the gradient (*Saunders et al., 2012*). These clusters represent nano-scale pockets of increased Pom1 concentration, but whether these clusters contribute to regulation of Cdr2 nodes is unknown. Here, we combine biochemical and microscopy approaches to show that the Pom1 gradient is comprised of these punctate clusters. Pom1 clusters are stable oligomers that bind and release the membrane with minimal lateral diffusion. These clusters bind more frequently at cell tips than at cell sides, resulting in a concentration gradient. A portion of Pom1 clusters in the medial cell cortex colocalize with Cdr2 nodes, and the ratio of Pom1 to Cdr2 at the medial cell cortex changes as a function of cell size. When glucose is restricted, Pom1 clusters redistribute to the medial cell cortex to prevent dramatic dysregulation of Wee1. Our work reveals that the Pom1-Cdr2-Wee1 signaling pathway is organized as a series of cortical clusters. The relative distribution of these clusters at the plasma membrane changes with cell size and glucose availability, thus relaying cell surface area to the core cell cycle machinery in a nutrient-controlled manner.

## Results

### The Pom1 gradient is formed by transient cortical clusters

We sought to examine the molecular dynamics of Pom1-Cdr2 signaling at the cell cortex. As a starting point, we imaged Pom1 by TIRF microscopy, which selectively excites fluorophores near the coverslip. Pom1 was tagged at the endogenous locus with the bright and photostable fluorophore mNeonGreen (mNG). Surprisingly, along the lateral cortex of interphase cells, Pom1-mNG was localized almost exclusively in discreet clusters, with no apparent diffuse signal (*Figure 1B*). In images from a single time-point, Pom1 clusters formed a noisy concentration gradient (*Figure 1B,D*). Time-averaging produced a smoother gradient dotted by occasional clusters (*Figure 1B,D*), similar to the gradient observed by confocal microscopy (*Figure 1—figure supplement 1A,B*). Pom1 clusters were highly dynamic (*Video 1*). They appeared on the cell sides with a frequency of ~2 $\mu m^{-2}s^{-1}$, and remained bound for less than 2 s on average (*Figure 1E,F*). During this brief cortical attachment, Pom1 clusters exhibited minimal diffusion which lacked directionality, and instead appeared to diffuse randomly in submicron patches of cortex (*Figure 1—figure supplement 2A,B,E*). Mean squared displacement (MSD) measurements were consistent with free diffusion of Pom1 clusters within submicron corrals (*Figure 1—figure supplement 2A,B,C,D*) (*Qian et al., 1991*). The measured diffusion coefficient (D = 0.134 ± 0.017 $\mu m^2/s$) was slow for a membrane-associated protein (*Knight et al., 2010*; *Weiß et al., 2013*), likely reflecting the large size and/or multiple membrane contacts for oligomeric Pom1 clusters.

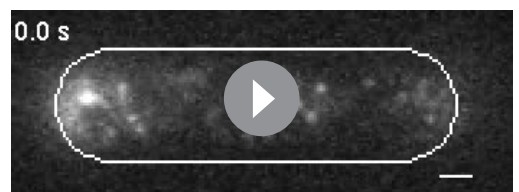

**Video 1.** Pom1-mNG forms clusters that transiently bind the cell cortex. TIRF microscopy of a cell expressing Pom1-mNG from the endogenous locus. Frames are 200 ms exposures from continuous acquisition. Scale bar is 1 μm.
DOI: https://doi.org/10.7554/eLife.46003.005

We next used 'head-on' TIRF microscopy to image Pom1 at cell tips. Similar to cell sides, Pom1-mNG localized at cell tips almost entirely in clusters (*Figure 1C*). We also used 'head-on' confocal imaging to confirm that Pom1 localization in clusters at tips was not an artifact of the TIRF approach (*Figure 1—figure supplement 1C*); a conclusion further supported by work using high resolution wide-field microscopy (*Dodgson et al., 2013*). Compared to cell sides, Pom1 clusters appeared twice as frequently at cell tips but with a similar cortical duration, again exhibiting minimal and non-directional diffusion away from their binding site (*Figure 1E,F*, *Figure 1—figure supplement 2E*). Fluorescence intensity of individual clusters at cell tips and cell sides were similar, although we detected a low number of brighter clusters at cell tips, likely representing multiple diffraction-limited clusters (*Figure 1—figure supplement 1C,D,E*). From these data, we conclude that the Pom1 gradient is formed by patterning the membrane binding frequency of cortical clusters along the long axis of the cell, rather than by diffusion of Pom1 molecules from the cell tips to the cell side.

## In vitro analysis shows Pom1 exists in large, stable clusters

Past work suggested that Pom1 structures assemble and disassemble at the cortex through oligo-merization of individual molecules diffusing in the plane of the membrane (*Saunders et al., 2012*). However, our high-speed (20 ms/frame) continuous TIRF videos suggested that Pom1 clusters bind and release the cortex as a pre-formed unit (*Figure 2—figure supplement 1A*). Consistent with this possibility, in the rare cases when we observed cell lysis events during TIRF imaging, Pom1 clusters remained intact in extruded cytoplasm (*Figure 2A*). This apparent stability while removed from the plasma membrane led us to test the existence and properties of Pom1 clusters in detergent cell extracts. Using TIRF microscopy, we observed clusters of Pom1-mNG in extracts made from Pom1-mNG cells, but not from untagged wild-type cells (*Figure 2B*). We next examined the size of these Pom1 clusters by velocity sucrose gradient sedimentation. By fractionating detergent cell extracts from *pom1-3xHA* cells, we found that most Pom1 exists in a large 60S complex, consistent with Pom1 clusters observed in cells and with previous results from gel filtration (*Bähler and Nurse, 2001*). Fractions 6–8 containing this peak were dialyzed and centrifuged on a second sucrose gradient. These pooled fractions again sedimented at 60S, indicating that they represent a stable complex (*Figure 2C*, *Figure 2—figure supplement 2A,B,C*).

Next, we tested the behavior of isolated Pom1 clusters on fluid artificial supported lipid bilayers (SLBs) (*Figure 2D*, *Figure 2—figure supplement 1B*). In this cell free-system, Pom1-mNG clusters bound and released SLB lipids with strikingly similar dynamics as in cells (*Figure 2E,F*, *Video 2*). MSD measurements for Pom1 clusters on SLBs revealed a diffusion coefficient similar to that measured in cells ($D_{cell}$ = 0.134 ± 0.017 $\mu m^2$/s vs $D_{SLB}$ = 0.168 ± 0.018 $\mu m^2$/s), but the relationship was linear over the lifetime of most particles (*Figure 2—figure supplement 3A,B,C,D*). These results indicate that whereas membrane diffusion of Pom1 clusters in cells is confined within microdomains, clusters can diffuse without confinement on synthetic bilayers. Dwell times were increased for the kinase-dead mutant Pom1(K728R)-mNG, which was previously shown to increase cortical Pom1 levels in cells (*Figure 2—figure supplement 1C*) (*Bähler and Pringle, 1998*; *Hachet et al., 2011*). Pom1 clusters did not colocalize with Cdr2 nodes in detergent extracts, suggesting they remain as separate structures (*Figure 2—figure supplement 1D*). Collectively, our TIRF microscopy and in vitro analysis of Pom1 clusters support a model whereby the majority of cellular Pom1 protein is contained within discreet, highly-stable oligomers that bind and release membranes with kinetics dictated by their catalytic activity in a pattern that generates a concentration gradient.

Formation of clusters could represent an intrinsic property of Pom1 protein, or alternatively might require additional cellular factors. To distinguish between these possibilities, we expressed and purified GST-Pom1 from bacteria, and then performed sucrose gradient centrifugation experiments. Recombinant GST-Pom1 sedimented in a low molecular weight peak unlike Pom1 clusters from cells (*Figure 2G*). Remarkably, purified GST-Pom1 was assembled into a cluster-sized high molecular weight complex by incubation with wild-type yeast detergent extract (*Figure 2G*). Complex formation was also induced by incubation of GST-Pom1 with *pom1Δ* cell extracts, meaning that additional cellular factors drive this assembly process (*Figure 2G*). Once assembled, Pom1 clusters are stable oligomeric complexes with intrinsic membrane-binding properties that can be reconstituted in vitro. Additional cellular regulatory proteins are likely to promote the binding of Pom1 clusters at cell tips, leading to the spatial gradient.

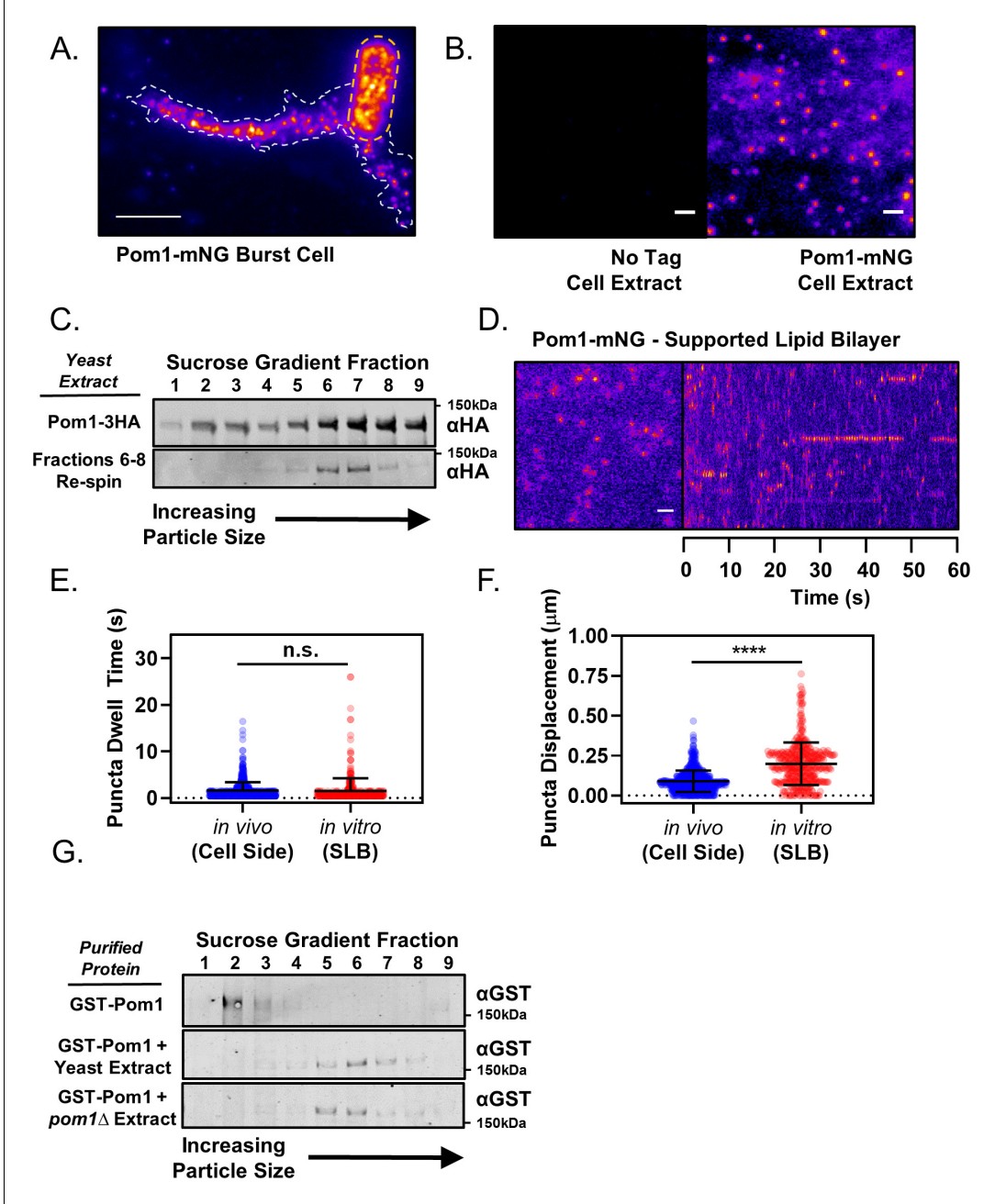

**Figure 2.** Pom1 clusters are stable structures that can be isolated in vitro. (**A**) TIRF microscopy image of Pom1-mNG clusters in the extruded cytoplasm (white dotted line) of a lysed cell (yellow dotted line). Scale bar 5 μm. (**B**) TIRF microscopy images of cell extracts prepared from wild-type (no tag) or Pom1-mNG cells. Images are 50 frame sum projections of continuous 200 ms time-lapse exposures. The two images were contrasted equally. Scale bars 1 μm. (**C**) Cytoplasmic extracts of *pom1-3HA* cells were subjected to velocity sucrose gradient sedimentation, and fractions were probed against the HA tag (upper blot). Fraction one corresponds to the top of the gradient and contains smaller structures; fraction nine corresponds to bottom of the gradient and contains larger structures. Fractions 6–8 were pooled, sucrose was removed by dialysis, and then the sample was subjected to a second identical round of sucrose gradient sedimentation and western blotting of the resulting fractions (lower blot). (**D**) TIRF microscopy of Pom1-mNG clusters from cytoplasmic extracts on supported lipid bilayers. Scale bar 1 μm. Left panel is single time point image. Right panel is kymograph taken from a line scan of time-lapse TIRF experiment. (**E**) Quantification of binding duration of Pom1-mNG clusters on supported lipid bilayers imaged by TIRF microscopy as in panel (**D**). Values are compared to cellular measurements of Pom1 clusters on cell sides (n.s., p=0.05954, n = 713 in vivo, 421 in vitro). (**F**) Quantification of life-time displacement of Pom1-mNG clusters diffusing on supported lipid bilayers imaged by TIRF microscopy as in panel (**D**). Values are compared to cellular measurements of Pom1 clusters on cell sides (****p<0.0001, n = 713 in vivo, 421 in vitro). For (**E**) and (**F**), statistical significance was tested using a Student's T-test. (**G**) Purified GST-Pom1 was subjected to sucrose gradient sedimentation and the fractions were probed

*Figure 2 continued on next page*

*Figure 2 continued*

against the GST tag (upper blot). Purified GST-Pom1 was also added to wild-type or *pom1Δ* cell extracts and incubated for 1 hr at 4 °C in the presence of ATP before velocity sucrose sedimentation and western blotting (bottom blots).

DOI: https://doi.org/10.7554/eLife.46003.006

The following figure supplements are available for figure 2:

**Figure supplement 1.** Controls and supporting in vitro analysis of Pom1 clusters.

DOI: https://doi.org/10.7554/eLife.46003.007

**Figure supplement 2.** Quantification of Pom1 sedimentation in velocity sucrose gradients and size standards.

DOI: https://doi.org/10.7554/eLife.46003.008

**Figure supplement 3.** Analysis of Pom1 cluster diffusion on supported lipid bilayers.

DOI: https://doi.org/10.7554/eLife.46003.009

## Polarity landmarks pattern cortical dynamics of Pom1 clusters to shape the gradient

The key role of clusters in forming the Pom1 gradient led us to reexamine the role of cell polarity landmark proteins Tea1 and Tea4. Both of these proteins localize at cell tips and are required for proper Pom1 gradient formation (*Hachet et al., 2011*). However, past studies have shown that Tea1 and Tea4 have distinct mechanistic roles: Tea4 is required for localization of Pom1 to the cortex, whereas Tea1 is required for enrichment of cortical Pom1 to the cell tip (*Hachet et al., 2011*). We first used sucrose gradient centrifugation to test if either protein is required for assembly of Pom1 into stable biochemical complexes. The sedimentation pattern of Pom1 clusters isolated from *tea1Δ* and *tea4Δ* cells in velocity sucrose gradients is unchanged from wild-type cells, suggesting that neither Tea1 nor Tea4 are required for assembly of Pom1 into stable biochemical complexes (*Figure 3A*) (*Bähler and Nurse, 2001*). Rather, these regulators are likely to act downstream of cluster assembly.

We next tested the localization and dynamics of Pom1-mNG in *tea1Δ* and *tea4Δ* cells. We confirmed that Pom1 was absent from the cell cortex in confocal micrographs of *tea4Δ* cells (*Figure 3—figure supplement 1A*, *Hachet et al., 2011*). Surprisingly, we did observe Pom1-mNG clusters throughout the cortex of *tea4Δ* cells by TIRF microscopy (*Figure 3B*). These binding events occurred with low frequency and short dwell times (*Figure 3C,D*, *Figure 3—figure supplement 1B,C,D*), which likely preclude their detection by confocal microscopy. In contrast, confocal micrographs of Pom1-mNG in *tea1Δ* cells confirmed even distribution around the entire cell cortex (*Figure 3—figure supplement 1A*, *Hachet et al., 2011*). In TIRF microscopy of *tea1Δ* cells, Pom1-mNG clusters bound the cell cortex with the same frequency at cell sides and cell tips (*Figure 3B,C,D*, *Figure 3—figure supplement 1B,C,D*) resulting in enrichment of cortical Pom1 but not a gradient. Thus, Tea1 and Tea4 cooperate to promote localized membrane binding but not assembly of Pom1 clusters. These spatial cues pattern the frequency and binding duration of Pom1 cluster cortical interaction along the long axis of the cell to generate a concentration gradient that emanates from cell tips.

## Pom1 clusters interact with Cdr2 nodes at the lateral cortex

It has been unclear when and where Pom1 interacts with its inhibitory target Cdr2, which localizes in cortical nodes positioned in the cell middle. Both Pom1 and Cdr2 are enriched at the cortex, and their cytoplasmic concentrations do not change with cell size (*Figure 4—figure supplement 1A,B,C, D,E*). Furthermore, Cdr2 nodes and Pom1 clusters do not colocalize in cytoplasmic extracts, suggesting the cytoplasm is not the location of their interaction (*Figure 2—figure supplement 1D*). Since Pom1 clusters appear to bind throughout the medial cell cortex, we used simultaneous two-color TIRF microscopy to test colocalization of Pom1 clusters and Cdr2 nodes in this region of overlap. We observed colocalization between Cdr2 nodes and some Pom1 clusters, as well as Pom1 clusters that bound to the cortex without encountering a Cdr2 node (*Figure 4A*). Thus, Pom1 clusters bind to the medial cortex in an apparently stochastic pattern that can generate overlap with Cdr2 nodes. These patterns of colocalization were apparent in both static images and time-lapse movies (*Figure 4A,B*). Similar results were obtained if different fluorophores were used to tag Pom1 and Cdr2, indicating that colocalization is unlikely an artifact of the fluorescent tags (*Figure 4—figure supplement 2A,B*). The frequency, dwell time, and displacement of Pom1 cortical clusters were

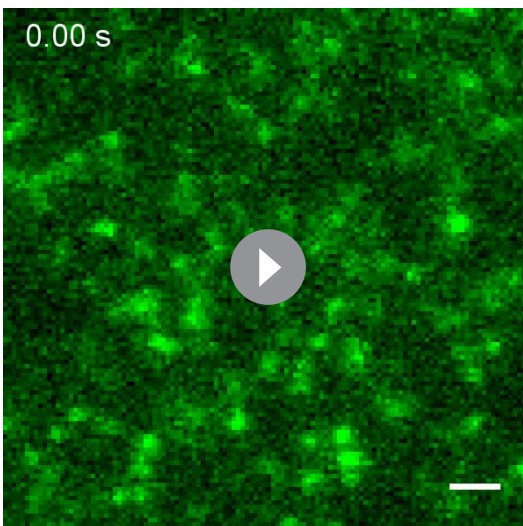

0.00 s

**Video 2.** Pom1-mNG clusters on synthetic supported lipid bilayers. TIRF microscopy of Pom1-mNG clusters in cytoplasmic extracts, binding and releasing artificial supported lipid bilayers composed of phosphatidylcholine and phosphatidylserine. Frames are 200 ms exposures with 200 ms interval between consecutive frames. Scale bar is 1 μm.
DOI: https://doi.org/10.7554/eLife.46003.010

unaffected in *cdr2Δ* cells (*Figure 4—figure supplement 3A,B,C*). These data suggest that Pom1 interacts with its substrate Cdr2 when Pom1 clusters associate with Cdr2 nodes at the medial cell cortex.

Phosphorylation by Pom1 prevents activation of Cdr2 in small cells, thereby contributing to cell size-dependent regulation of Wee1 and mitotic entry (*Allard et al., 2018*). To place this genetic pathway in the cellular context of Pom1 clusters and Cdr2 nodes, we analyzed how these structures accumulate and colocalize as cells grow. Intriguingly, the total number of both Cdr2 nodes and Pom1 clusters detected along entire cell sides in TIRF images were equivalent for cells of a given size (*Figure 4C,D*), and both structures show a similar size-dependent doubling. We next restricted our analysis to a 2 μm x 2 μm square ROI positioned at the cell middle, where Cdr2 nodes concentrate. The local density of Cdr2 nodes in this region increased as a function of cell size, consistent with past studies (*Deng and Moseley, 2013*; *Pan et al., 2014*). In contrast, the density of Pom1 clusters in this region was largely independent of cell size (*Figure 4E*). In both TIRF and confocal images, the concentration of Pom1 protein and Pom1 clusters decreased slightly as cells increase in size, but this trend was

dwarfed by the cell size-dependent increase in Cdr2 node density (*Figure 4E*, *Figure 4—figure supplement 3D*). Thus, as cells grow larger, the ratio of Pom1 to Cdr2 in the medial cell cortex changes to favor Cdr2 because the densities of these two structures scale differently with cell size (*Figure 4G*).

We next tested how this density scaling affects colocalization of Pom1 clusters and Cdr2 nodes as cells grow. Due to the increasing density of Cdr2 nodes, we observed an increased number of colocalized Pom1 clusters and Cdr2 nodes at cell sides as cells increase in size (*Figure 4F*, *Figure 4—figure supplement 3E*). The increased colocalization, combined with the constant density of Pom1 clusters, means that there are less 'free' Pom1 clusters as cells increase in size. Perhaps more importantly, the number of 'free' Cdr2 nodes increased as a function of cell size (*Figure 4—figure supplement 3F*). These free Cdr2 nodes are not occupied by an inhibitory Pom1 cluster, and thus have increased potential to promote mitotic entry by inhibiting Wee1. In this manner, Pom1 sets an inhibitory threshold that must be overcome by an increase in Cdr2 node density. The inhibitory threshold decreases slightly as cells grow, and functions most effectively in small cells, where Pom1 was previously shown to inhibit downstream signaling to Wee1 (*Allard et al., 2018*).

## Control of Pom1 cluster levels in the medial cell cortex

Our results suggest that Pom1 sets an inhibitory threshold that opposes Cdr2 activation until cells reach an appropriate size for division. We hypothesized that the strength of the Pom1 inhibitory threshold could be tuned by altering the concentration of Pom1 cortical clusters in the cell middle. To increase the abundance of Pom1 clusters in the cell middle, we grew Pom1-mNG cells in low glucose media. In low glucose conditions, the fission yeast microtubule cytoskeleton depolymerizes, which leads to Pom1 redistribution to the lateral cell cortex where it delays mitotic entry (*Kelkar and Martin, 2015*). In time-lapse TIRF microscopy experiments, we observed increased numbers of discreet Pom1 clusters at the lateral cell cortex under low glucose conditions (*Figure 5A*). These clusters bound to the lateral cortex more frequently than under normal glucose conditions, but the average dwell time was unaffected (*Figure 5B,C*). This increased on-rate leads to accumulation of Pom1 clusters at the medial cell cortex.

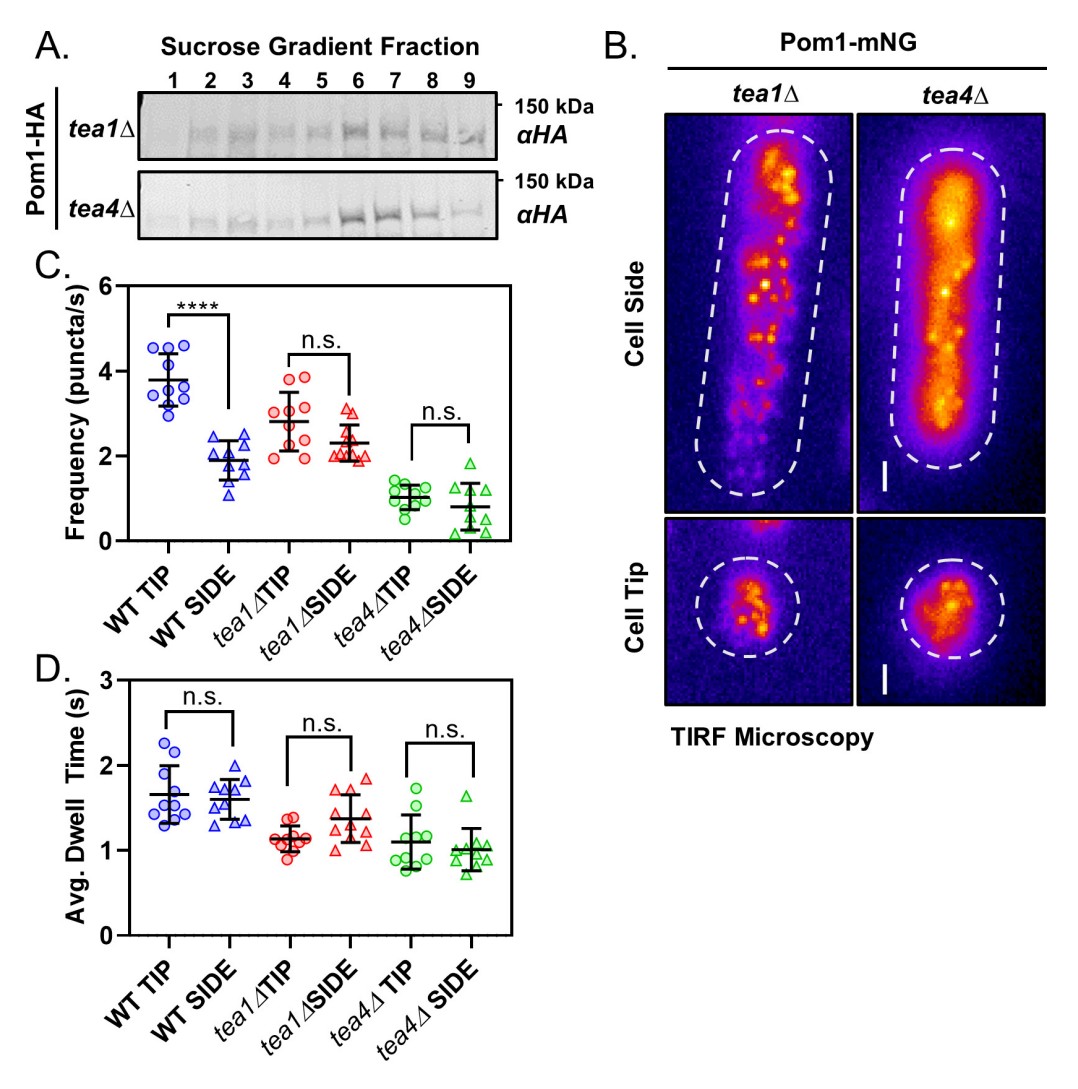

**Figure 3.** Polarity landmarks pattern cortical dynamics of Pom1 clusters to shape the gradient. (**A**) Western blots of sucrose gradient fractions after sedimentation analysis as in *Figure 2C* of extracts prepared from *tea1Δ* or *tea4Δ* cells expressing Pom1-3HA. (**B**) TIRF micrographs of Pom1-mNG localization at the tip or side cortex in *tea1Δ* or *tea4Δ* cells. Scale bar 1 μm. (**C**) The mean binding frequency of Pom1-mNG clusters at the tips and sides of wild-type, *tea1Δ* or *tea4Δ* cells (*n* = 10 cells, 7–170 traces/cell). (**D**) The mean dwell time of Pom1-mNG clusters at the tips and sides of wild-type, *tea1Δ* or *tea4Δ* cells. For panels C-D, wild-type data are replotted from main (*Figure 1*). Each data point represents a single cell mean, and line and error bars represent mean and standard deviation of all cells (*n* = 10 cells, 7–170 traces/cell). Comparisons are 1-way ANOVA with Tukey's multiple comparisons tests. (****) indicates p<0.0001, (n.s.) indicates p>0.05. See *Figure 3—figure supplement 1D* for all results of statistical analysis.

DOI: https://doi.org/10.7554/eLife.46003.011

The following figure supplement is available for figure 3:

**Figure supplement 1.** Supporting analysis of Pom1 clusters in polarity mutants.

DOI: https://doi.org/10.7554/eLife.46003.012

We tested how increased numbers of Pom1 clusters at the lateral cell cortex affect Cdr2 nodes using TIRF microscopy. In low glucose conditions, the number of Cdr2 nodes measured per cell was largely unchanged. However, we observed a significant reduction in the number of Cdr2 molecules per node, and in the overall concentration of Cdr2 at the medial cortex (*Figure 5D,E,F*, *Figure 5—figure supplement 1A,B,C*). Low glucose did not affect Cdr2 nodes in *pom1Δ* cells, so these effects

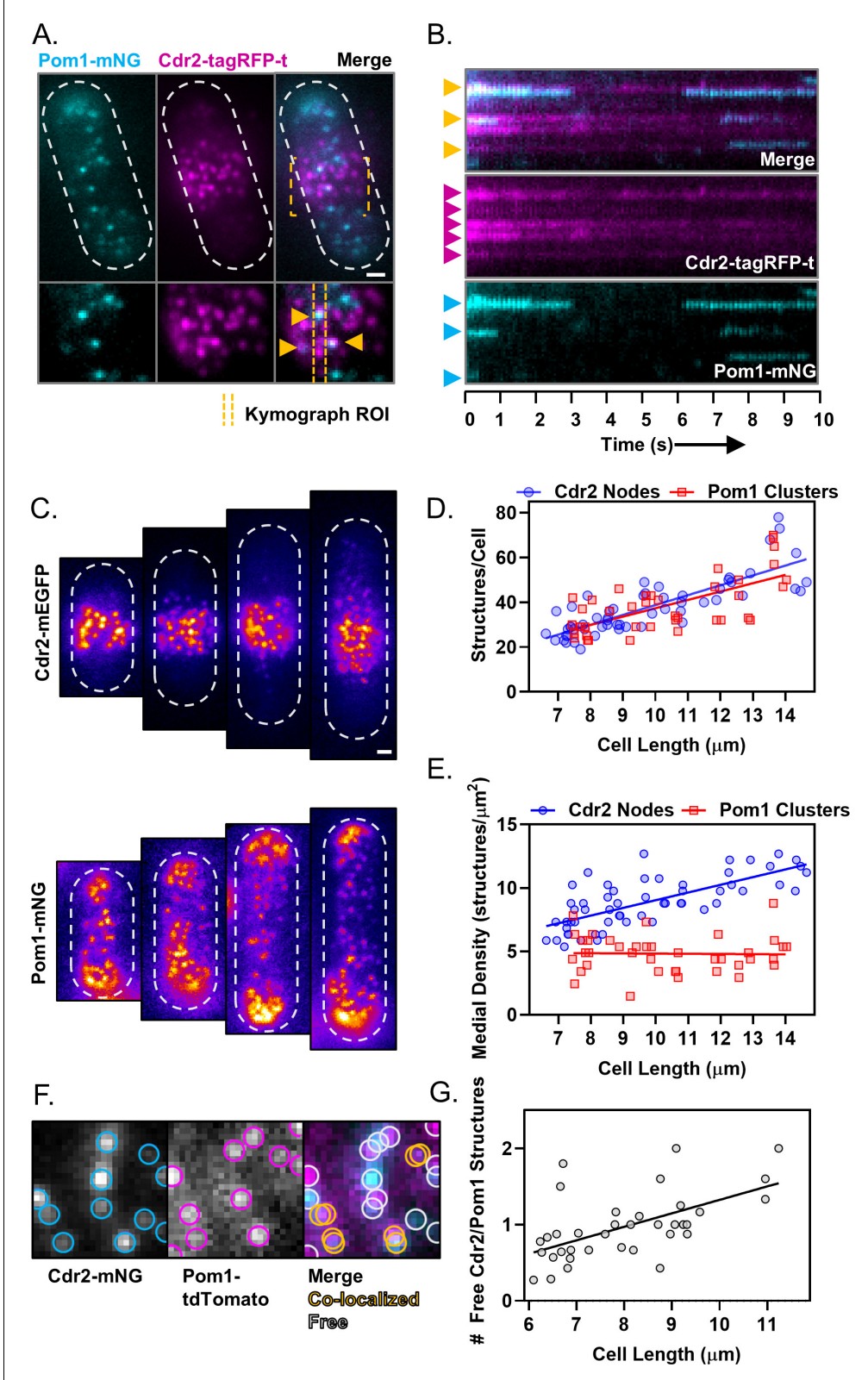

**Figure 4.** Size-scaling of Pom1 clusters and Cdr2 nodes at the lateral cell cortex. (**A**) A subset of Pom1 clusters colocalize with Cdr2 nodes. Panels are TIRF microscopy images of cells expressing Pom1-mNG and Cdr2-tagRFP-t. Yellow dashed brackets outline the ROI of the lower zoomed panels. Orange arrows point to Pom1 clusters colocalized with Cdr2 nodes. Scale bar is 1 μm. (**B**) Kymographs generated using the ROI indicated in panel A. Cyan arrows indicate prominent Pom1 clusters, magenta arrows indicate prominent Cdr2 nodes, and orange arrows indicate colocalization. (**C**)
*Figure 4 continued on next page*

*Figure 4 continued*

Localization of Cdr2-mEGFP (upper panels) or Pom1-mNG (lower panels) in representative cells of increasing size, imaged by TIRF microscopy. Scale bar is 1 μm. (D) Total number of Cdr2-mEGFP nodes (blue circles) or Pom1-mNG clusters (red squares) measured per cell. Quantification is limited to clusters detected and resolvable in the TIRF illumination field. The slopes of the corresponding linear regressions are not significantly different (p=0.3757, *n* > 40 cells). (E) Density of Cdr2-mEGFP nodes (blue circles) or Pom1-mNG clusters (red squares) in 2 × 2 μm square ROIs at the cell middle, counted using TIRF microscopy. The slopes of the corresponding linear regressions are significantly different (p<0.0001, *n* > 40 cells). (F) Example of colocalization analysis for Pom1 clusters (magenta ROI) and Cdr2 nodes (cyan ROI) (left panels). Colocalized structures are marked with overlapping yellow ROIs, whereas non-colocalizing structures are marked with gray ROIs (right panel, gray circles). (G) Ratio of free Cdr2 nodes to free Pom1 clusters, plotted as a function of cell size. The slope of the linear regression is positive and significantly non-zero (p=0.0002, $R^2$ = 0.33, *n* = 36 cells).

DOI: https://doi.org/10.7554/eLife.46003.013

The following figure supplements are available for figure 4:

**Figure supplement 1.** Quantification of Pom1 and Cdr2 concentration in different cellular regions.
DOI: https://doi.org/10.7554/eLife.46003.014

**Figure supplement 2.** Colocalization of Cdr2 nodes and Pom1 clusters using alternative fluorophore pair.
DOI: https://doi.org/10.7554/eLife.46003.015

**Figure supplement 3.** Supporting analysis of Cdr2 nodes and Pom1 clusters at the medial cell cortex.
DOI: https://doi.org/10.7554/eLife.46003.016

---

are Pom1-dependent (*Figure 5D,E,F*). Thus, Pom1 redistribution to the lateral cortex under low glucose is required to induce partial disassembly of each Cdr2 node. We quantified how this redistribution effect scales with glucose levels. As glucose was reduced, Pom1 concentration decreased at cell tips and increased at cell sides (*Figure 6A,C,D*). Pom1 concentration at the lateral cortex therefore serves as a cellular indicator of glucose availability. In contrast, Cdr2 concentration increased at cell tips and decreased at cell sides (*Figure 6B,C,D*). The concentrations of Pom1 and Cdr2 at cell sides were nearly perfectly anti-correlated (Pearson's r = −0.9352, p=0.0006) (*Figure 6E,F*). We conclude that the increasing frequency of Pom1 cluster binding at the lateral cell cortex leads to a reduced number of Cdr2 molecules per node.

## Pom1 disrupts Wee1 bursting under glucose deprivation

We next examined the downstream effects of altered Pom1-Cdr2 signaling in low glucose by using TIRF microscopy to monitor the previously described bursts of Wee1-mNG localization at Cdr2 nodes (*Allard et al., 2018*). In wild-type cells, both the frequency and duration of Wee1 bursts increased linearly with cell size, consistent with increased inhibition of Wee1 as cells grow (*Figure 7A,B*, *Supplementary file 1*, *Supplementary file 2*). In low glucose, the frequency of Wee1-mNG bursts still scaled with cell size as in normal glucose. However, the duration of each burst was uniformly short and independent of cell size (*Figure 7A,B*, *Supplementary file 1*, *Supplementary file 2*). Therefore, glucose modulates how long Wee1 molecules stay at inhibitory Cdr2 nodes. These same properties were seen for Wee1-mNG bursts in kinase-dead *cdr2(E177A)* mutant cells grown in normal glucose (*Figure 7—figure supplement 1A,B*, *Supplementary file 1*, *Supplementary file 2*). These results indicate that Cdr2 kinase activity is required to retain Wee1 at nodes, but not for the initial binding event. Further, they demonstrate that low glucose phenocopies the kinase-dead *cdr2(E177A)* mutant, consistent with Pom1-dependent inhibition of Cdr2 kinase activity.

These results suggested that reduced Wee1 bursting in low glucose might require Pom1 redistribution. We tested this idea by examining the glucose dependency of Wee1 localization to nodes in *pom1Δ* cells. In *pom1Δ* cells growing under high glucose, the Wee1 burst frequency scales with cell size but Wee1 burst duration is uniformly high and independent of cell size (*Figure 7A,C*, *Supplementary file 1*, *Supplementary file 2*). Unlike wild type cells, which suppress Wee1 bursts in low glucose, *pom1Δ* cells show a striking increase in Wee1 bursts under low glucose. This increase is not caused by burst duration, which was independent of both cell size and glucose concentration in *pom1Δ* cells (*Figure 7C*, *Supplementary file 1*, *Supplementary file 2*). Rather, we measured increased Wee1-mNG burst frequency in low glucose *pom1Δ* cells (*Figure 7A,C*, *Supplementary file 1*, *Supplementary file 2*). The reason for this increase is unknown, and suggests the existence of additional glucose-regulated pathways that operate on Cdr2-Wee1 signaling.

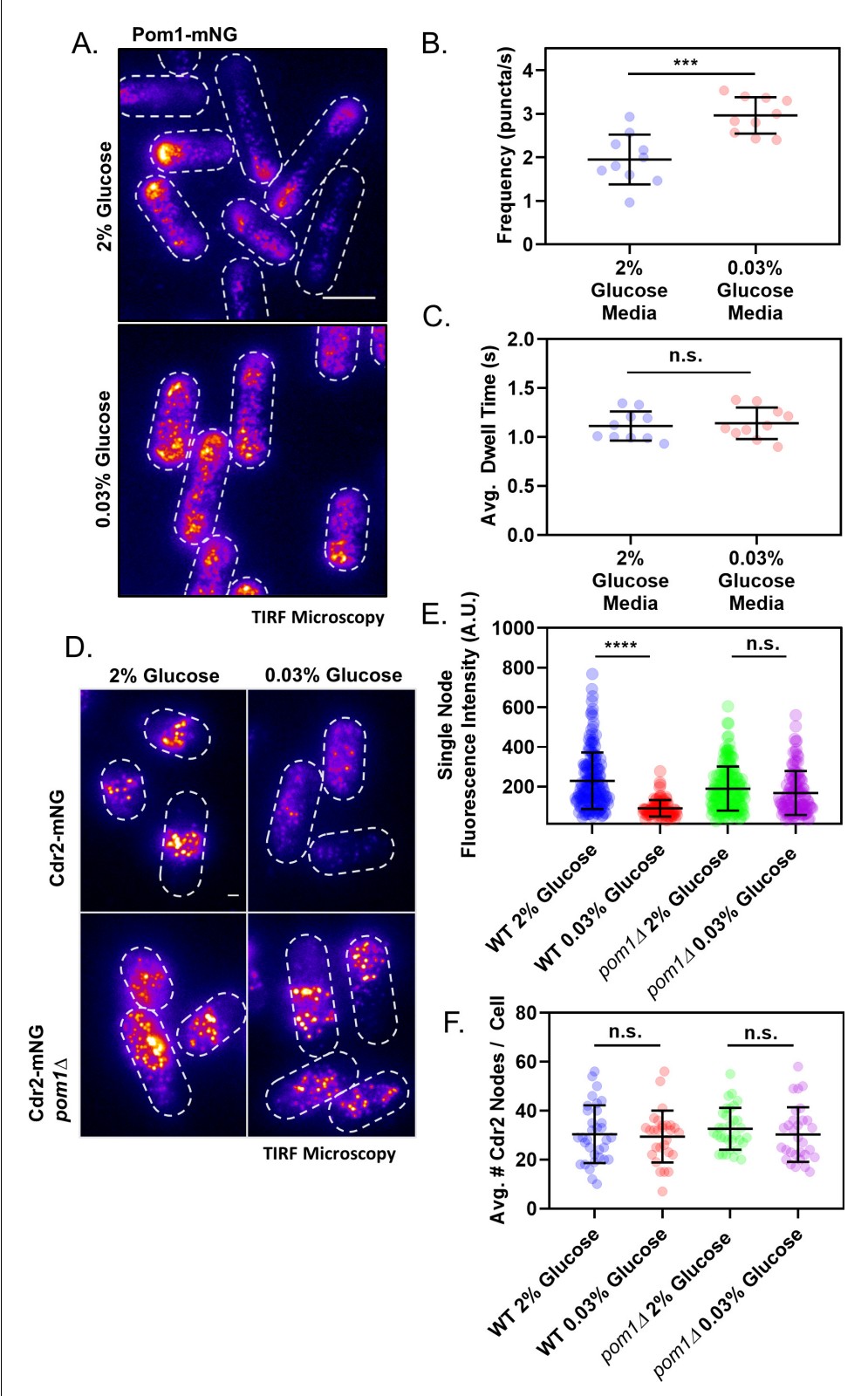

**Figure 5.** Redistribution of Pom1 clusters to the lateral cortex disrupts Cdr2 nodes during glucose restriction. (**A**) Localization of Pom1-mNG in cells grown in either high (2%) or low (0.03%) glucose media. Images were collected using TIRF microscopy. Scale bar, 5 μm. (**B**) Comparison of the binding frequency of Pom1 clusters at the lateral cell cortex in cells grown under high and low glucose (***p=0.0003, *n* = 10 cells, 32–113 traces/cell). Statistical significance was tested using a Student's T-test. (**C**) Comparison of the binding dwell time of Pom1 clusters at the lateral cell cortex in cells grown
*Figure 5 continued on next page*

*Figure 5 continued*

under high and low glucose (n.s., p=0.6833, *n* = 10 cells, 32–113 traces/cell). Statistical significance was tested using a Student's T-test. (D) Localization of Cdr2-mNG in wild-type or *pom1Δ* cells grown in either normal (2%) or low (0.03%) glucose media. Images were collected using TIRF microscopy. Scale bar, 1 μm. (E) Comparison of the fluorescence intensity of individual Cdr2-mNG nodes in wild-type or *pom1Δ* cells grown in either normal (2%) or low (0.03%) glucose media. Low glucose induces partial node disassembly in wild-type cells (****p<0.0001, *n* = 92–169 nodes from >5 cells) but not in *pom1Δ* cells (n.s., p=0.4015, *n* = 113–156 nodes from >5 cells). Measurements were taken from Airyscan Super-Resolution confocal micrographs. (F) Quantification and comparison of the total number of Cdr2-mNG nodes visible in TIRF micrographs of wild-type or *pom1Δ* cells grown in either normal (2%) or low (0.03%) glucose media. There is no significant difference in any condition (p>0.05, *n* = 28–33 cells). Statistical significance was tested using a one-way ANOVA.

DOI: https://doi.org/10.7554/eLife.46003.017

The following figure supplement is available for figure 5:

**Figure supplement 1.** Redistribution of Pom1 clusters to the lateral cortex disrupts Cdr2 nodes during glucose restriction, supporting data.

DOI: https://doi.org/10.7554/eLife.46003.018

Wee1 localization at Cdr2 nodes leads to its inhibitory phosphorylation by Cdr2 and the related kinase Cdr1. To test how our microscopy results connect with Wee1 phosphorylation status, we analyzed phosphorylation-dependent shifts of Wee1 migration using western blots (*Figure 7D*). In high glucose, Wee1 migrates as a smear of phosphorylated isoforms (*Allard et al., 2018*; *Lucena et al., 2017*). The upper, hyperphosphorylated forms are increased in *pom1Δ* cells but absent in *cdr2Δ* cells. This hyperphosphorylated Wee1 is lost in wild-type cells grown in low glucose, consistent with reduced bursts of Wee1 localization to nodes. In contrast, Wee1 appears even more hyperphosphorylated in *pom1Δ* cells grown under low glucose. This result suggests that Pom1 prevents phospho-inactivation of Wee1 in response to low glucose. These combined experiments support a model where Pom1 redistribution to the lateral cell cortex inactivates Cdr2 nodes to relieve inhibition of Wee1 under low glucose conditions.

## Discussion

In this study, we have shown that the Pom1 concentration gradient is generated by oligomeric Pom1 clusters that rapidly bind and release from the plasma membrane. These clusters bind more frequently at cell tips versus cell sides in high glucose media, while their binding rate is increased at cell sides in low glucose media. At the medial cell cortex, these clusters overlap with their inhibitory target Cdr2, which localizes in static oligomeric nodes. More Cdr2 nodes are free from Pom1 inhibition as cells grow larger due to different density scaling of these two structures at the medial cell cortex. Our TIRF-based colocalization experiments on these structures were limited to short periods of time to avoid photobleaching, partly caused by the imaging conditions needed to observe these highly dynamic Pom1 clusters. Nevertheless, Pom1 clusters colocalized with Cdr2 nodes even over these short timescales. Given the high frequency of Pom1 cluster binding, we expect that many more colocalization events occur between Cdr2 nodes and Pom1 clusters during the course of a full cell cycle, with the potential for Pom1 clusters to visit each Cdr2 node multiple times. Thus, these rapid dynamics of Pom1 clusters interacting with Cdr2 nodes may be integrated in time throughout the cell cycle. It appears likely that the most critical temporal window for Pom1-Cdr2 interactions occurs during early G2, as *pom1Δ* show defects in Cdr2-Wee1 signaling specifically in small cells (*Allard et al., 2018*). It will be interesting to determine how these rapid interactions in small cells are integrated into the mitotic entry decision, which occurs later in the cell cycle when cells are longer.

### Molecular clusters are widespread in signal transduction

The Pom1-Cdr2-Wee1 signaling pathway appears to function entirely within the confines of oligomeric protein clusters at the plasma membrane. The development of imaging technologies with increased signal-to-noise and spatial and temporal resolution has enabled the discovery of protein clustering as a paradigm in signaling. As a result, a growing number of signaling events and proteins have been found in similar clustered structures that are referred to as nanodomains, clusters, nanoclusters, and nodes, among other names. Such structures have been shown to operate in polarization of the *C. elegans* zygote, signal transduction nanodomains at the plant plasma membrane, Ras isoform-specific signaling, bacterial chemotactic receptors, and both B-cell and T-cell receptors

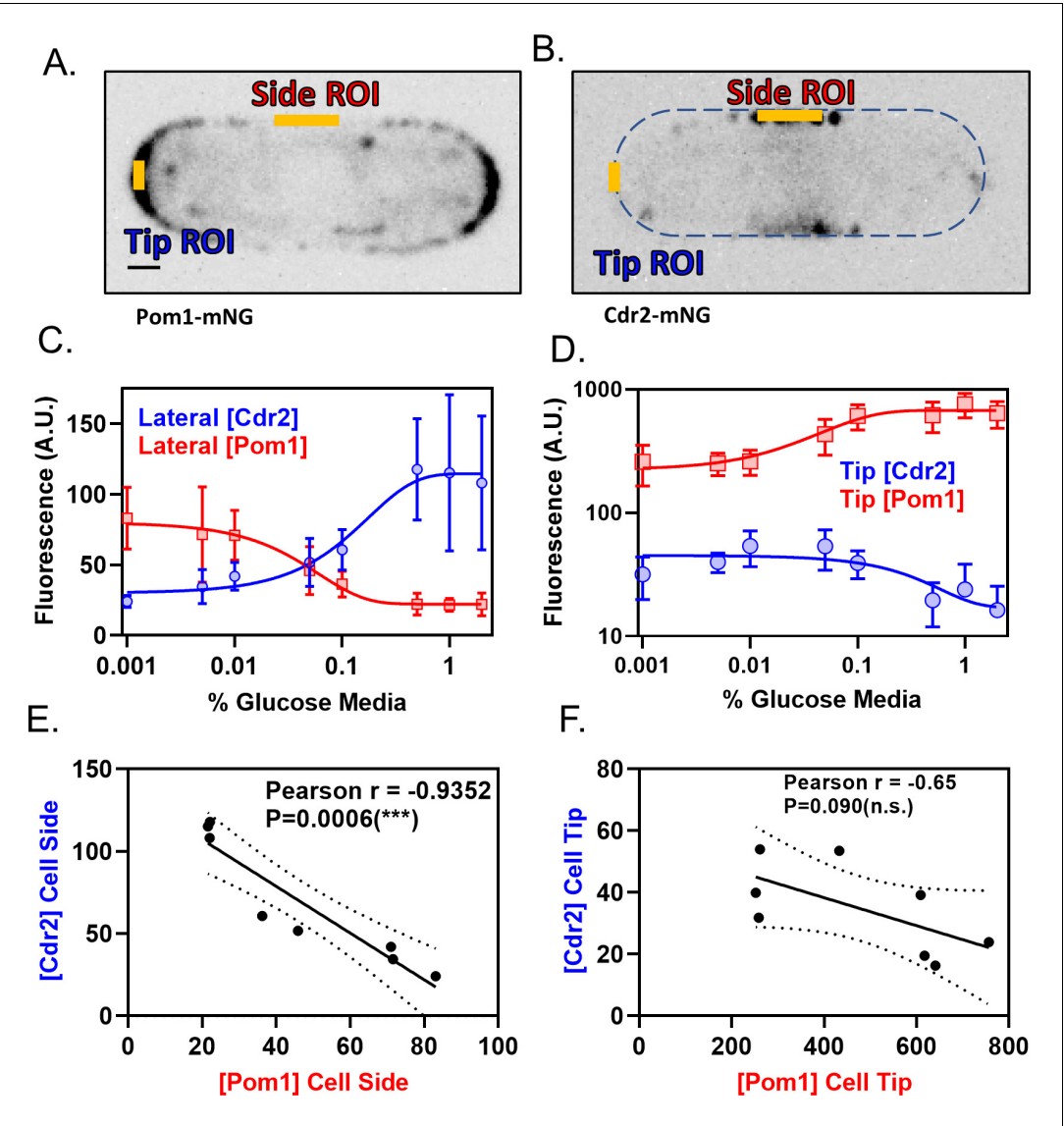

**Figure 6.** Pom1 and Cdr2 concentrations at the lateral cell cortex depend on glucose availability and are anti-correlated. (A–B) Localization of Pom1-mNG (panel A) and Cdr2-mNG (panel B) in confocal micrographs at middle cell focal planes. Regions where line-scans were used to measure fluorescence intensity are marked at the tip and side (orange lines). Scale bar, 1 μm. (C) Concentration of Pom1 (red squares) or Cdr2 (blue circles) at cell sides in a range of glucose concentrations, measured using 'Side ROI' as in panels A and B ($n > 10$ cells/concentration). (D) Concentration of Pom1 (red squares) or Cdr2 (blue circles) at cell tips in a range of glucose concentrations, measured using 'Tip ROI' as in panels A and B ($n > 10$ cells/concentration). (E) Correlation of Cdr2 vs Pom1 concentrations at cell sides in each glucose concentration from panel C. Concentrations are anticorrelated across all tested media glucose concentrations (p=0.0006, Pearson $r = -0.9352$, $n = 8$ concentrations). (F) Correlation of Cdr2 vs Pom1 concentrations at cell tips in each glucose concentration from panel D. Concentrations show weak anticorrelation across the tested media glucose concentrations, but the correlation is not statistically significant (p=0.09, Pearson $r = -0.65$, $n = 8$ concentrations).
DOI: https://doi.org/10.7554/eLife.46003.019

(*Bray et al., 1998*; *Dickinson et al., 2017*; *Douglass and Vale, 2005*; *Duke and Bray, 1999*; *Falke, 2002*; *Gronnier et al., 2017*; *Liu et al., 2016*; *Maity et al., 2015*; *Munro, 2017*; *Rodriguez et al., 2017*; *Wang et al., 2017*; *Zhou and Hancock, 2015*). These T cell receptor clusters enrich downstream kinases while excluding phosphatases (*Su et al., 2016*), a mechanism that could relate to concentrated clusters of the kinases Pom1 and Cdr2. In addition, theoretical work

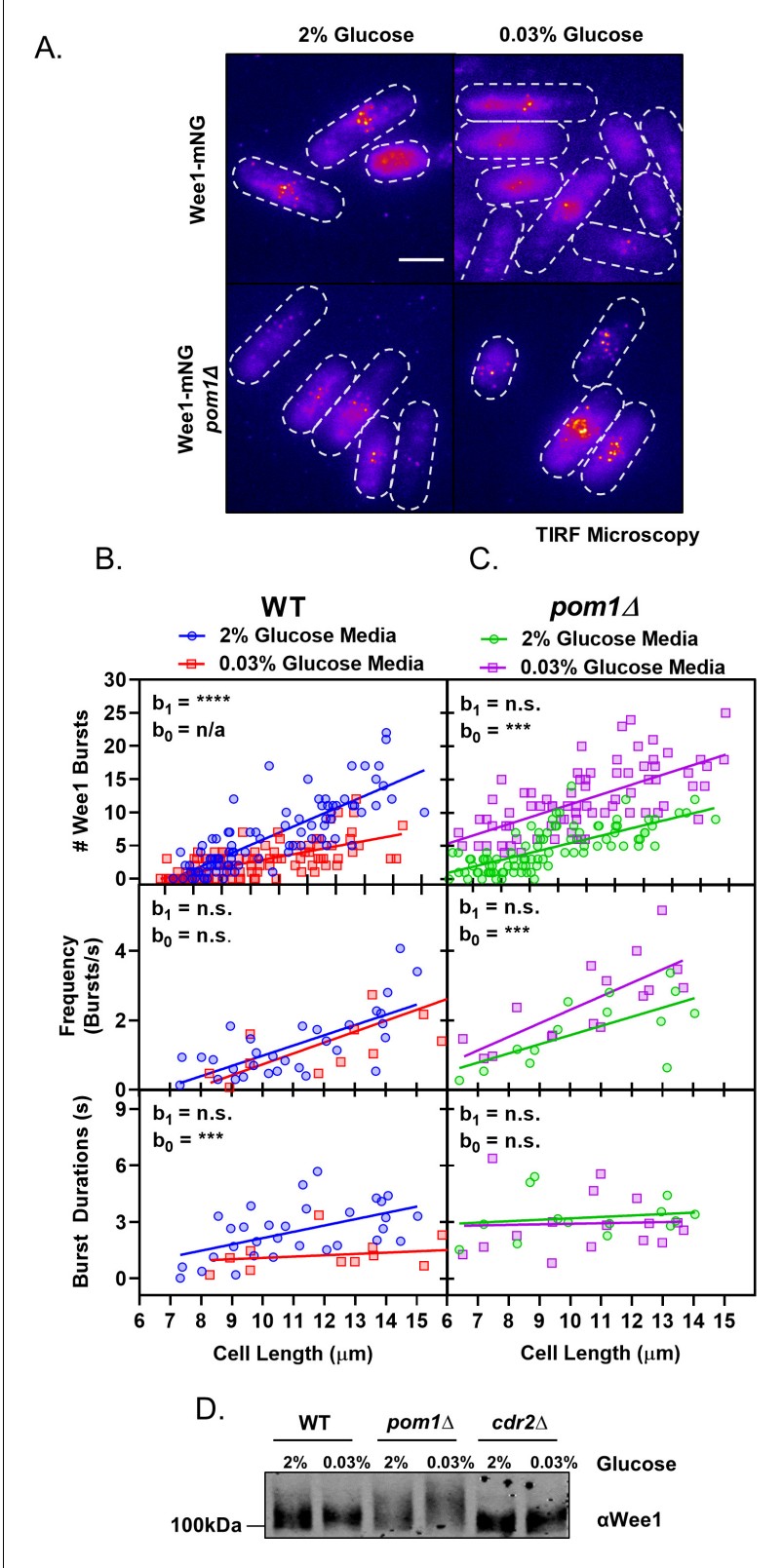

**Figure 7.** Pom1 redistribution under glucose restriction disrupts Wee1 regulation at cortical nodes. (**A**) Localization of Wee1-mNG in wild-type or *pom1Δ* cells grown in either high (2%) or low (0.03%) glucose media. Images were collected using TIRF microscopy. Scale bar, 5 μm. (**B**) Quantification of Wee1 bursting kinetics in wild-type cells grown under high (2%, blue circles) and low (0.03%, red squares) glucose conditions. The top panel

*Figure 7 continued on next page*

*Figure 7 continued*
is a plot of the total number of Wee1 bursts as a function of cell length, counted in single time point TIRF micrographs. The middle panel is a plot of the frequency of Wee1 bursts as a function of cell length. The bottom panel is a plot of Wee1 burst duration as a function of cell length. (C) Quantification of Wee1 bursting kinetics in *pom1Δ* cells grown under normal (2%, green circles) and low (0.03%, magenta squares) glucose conditions, as in panel B. For (B–C), insets represent whether the best fit values of the slope ($b_1$) and/or Y-intercept ($b_0$) of the two linear regressions are different at the 95% confidence level. See *Supplementary file 1* and *Supplementary file 2* for parameters of linear regressions and full statistical comparisons. (D) Western blot of Wee1 in wild-type, *pom1Δ*, and *cdr2Δ* cells grown in high (2%) or low (0.03%) glucose media showing phosphorylation dependent band-shifts.

DOI: https://doi.org/10.7554/eLife.46003.020
The following figure supplement is available for figure 7:

**Figure supplement 1.** Wee1 bursting in low glucose phenocopies a kinase-dead Cdr2 mutant.
DOI: https://doi.org/10.7554/eLife.46003.021

---

has suggested that clustering can reduce noise by generating reaction bursts (*Kalay et al., 2012*), which have the capacity to overcome inhibitory thresholds, such as those imparted by phosphatases, more effectively than a system driven by gradual accumulation of signal. Insight into the functional role of clustering for Pom1-Cdr2-Wee1 will require additional in vivo analysis combined with in vitro reconstitution. We have shown that Pom1 clusters remain stable through biochemical enrichment, and isolated clusters exhibit the same membrane-binding properties in vitro as in cells. Past work has demonstrated Cdr2 nodes also remain stable through biochemical fractionation (*Allard et al., 2018*), providing the necessary tools for future dissection of the pathway in a reconstituted system.

## A new model for cortical gradient formation

Our work has shown that Pom1 clusters bind to the cell cortex with different frequencies along the long axis of the cell, resulting in a concentration gradient. Past work suggested single Pom1 molecules (or small oligomers) bind to the membrane at tips, and then diffuse away from the tip before dissociating (*Hachet et al., 2011*; *Saunders et al., 2012*). Clusters were proposed to dynamically assemble and disassemble at the cortex from this diffusing population (*Saunders et al., 2012*). Our findings suggest a new model for gradient formation, which is distinct from this previous model in four ways. First, we found that clusters themselves form a concentration gradient that emanates from the cell tips. Second, our combined imaging and biochemical approaches demonstrated that clusters are stable complexes that bind and rapidly release from the membrane. Third, we did not observe extensive diffusion by complexes bound to the membrane. It remains possible that smaller, diffusing Pom1 complexes or single molecules were not detected by our imaging approach, but we favor a model where diffusion is limited to short, non-directional movements that do not contribute to gradient formation. Fourth, the key step in gradient formation is a positional system that increases the on-rate for Pom1 clusters at cell tips versus the cell middle. Concentration gradients are found across vast size scales in biology, so the mechanism of gradient formation by Pom1 clusters may have broad implications. It is also important to note that the spatial gradient of on-rates for Pom1 clusters binding to the cortex may depend on a concentration gradient of specific lipids in the plasma membrane. In support of this hypothesis, Pom1 binds to phosphatidylserine in vitro (*Hachet et al., 2011*), and this lipid is enriched at the tips of growing fission yeast cells (*Haupt and Minc, 2017*). Thus, further investigation of these lipids and their localization dynamics could reveal additional layers of this morphogen-like gradient.

We found that the polarity landmarks Tea1 and Tea4 provide the positional information for Pom1 clusters to form a concentration gradient. Past work has shown that Pom1 localizes homogenously throughout the cortex in *tea1Δ* cells, and does not localize to the cortex in *tea4Δ* cells (*Bähler and Pringle, 1998*; *Hachet et al., 2011*). Based on these and other results, Tea1 was proposed to recruit Tea4 to cell tips, where it promotes cortical loading of Pom1 molecules. Several results suggest that additional mechanisms may contribute to gradient formation. For example, we found that Pom1 clusters still bind to the cortex in *tea4Δ* cells but for very short durations. Past work has also shown that Tea1 and Tea4 form cortical clusters that co-localize at cell tips, but these Tea1-Tea4 clusters do not co-localize with Pom1 clusters (*Dodgson et al., 2013*). Thus, Tea1-Tea4 may pattern binding of Pom1 clusters to the cortex through an additional mechanism. For example, Tea1 and Tea4 have

been shown to assemble sterol-enriched lipid domains at cell tips when cells initiate polarization following exit from starvation (*Makushok et al., 2016*). Sterols and phosphatidylserine are thought to form a common membrane domain in fission yeast, suggesting that these proteins could act through the recruitment and organization of lipids to generate cortical domains permissive for Pom1 cluster binding (*Haupt and Minc, 2017*; *Makushok et al., 2016*).

## Adjustable Pom1 gradients regulate mitotic entry

We propose an integrated mechanism for fission yeast cell size sensing in which Cdr2 acts as an important sensor of cell size that is dynamically antagonized by cortical Pom1 (*Figure 8*). Pom1 clusters bind to the medial cell cortex in a manner that can overlap with Cdr2 nodes. These dynamic Pom1 clusters and static Cdr2 cortical nodes suggest a system that monitors surface expansion, consistent with the notion that fission yeast cells monitor surface area as the primary determinant of size at division (*Facchetti et al., 2019*; *Pan et al., 2014*). As cells grow, the density of Pom1 clusters in the cell middle decreases slightly due to their binding preference at cell tips. In contrast, the density of Cdr2 nodes doubles to overcome this inhibitory Pom1 threshold. The density of Pom1 clusters and the resulting inhibitory threshold can be modulated by glucose availability, consistent with past work (*Kelkar and Martin, 2015*). As fission yeast cells begin to starve due to depletion of nutrients such as glucose, their length at division progressively decreases and the length of the cell cycle increases (*Pluskal et al., 2011*; *Yanagida et al., 2011*). The Pom-Cdr2 network has a prominent role in this nutrient modulation, suggesting that it may represent a key physiological role for the pathway (*Kelkar and Martin, 2015*). We found that the size dependence of Wee1 localization bursts to Cdr2 nodes, which facilitate Wee1 inhibition, was dampened in low glucose conditions. Re-localization of Pom1 clusters to the medial cortex mediates this response. However, for *pom1Δ* mutants in low glucose, Wee1 localization to nodes exceeds even wild type cells in high glucose. This result indicates that additional mechanisms contribute to Wee1 regulation in low glucose, for example Wee1 nuclear transport could be altered under these conditions. Thus, Pom1-Cdr2 signal transduction serves two functions: (1) Under steady-state growth conditions, Pom1 and Cdr2 cooperate to measure cell size

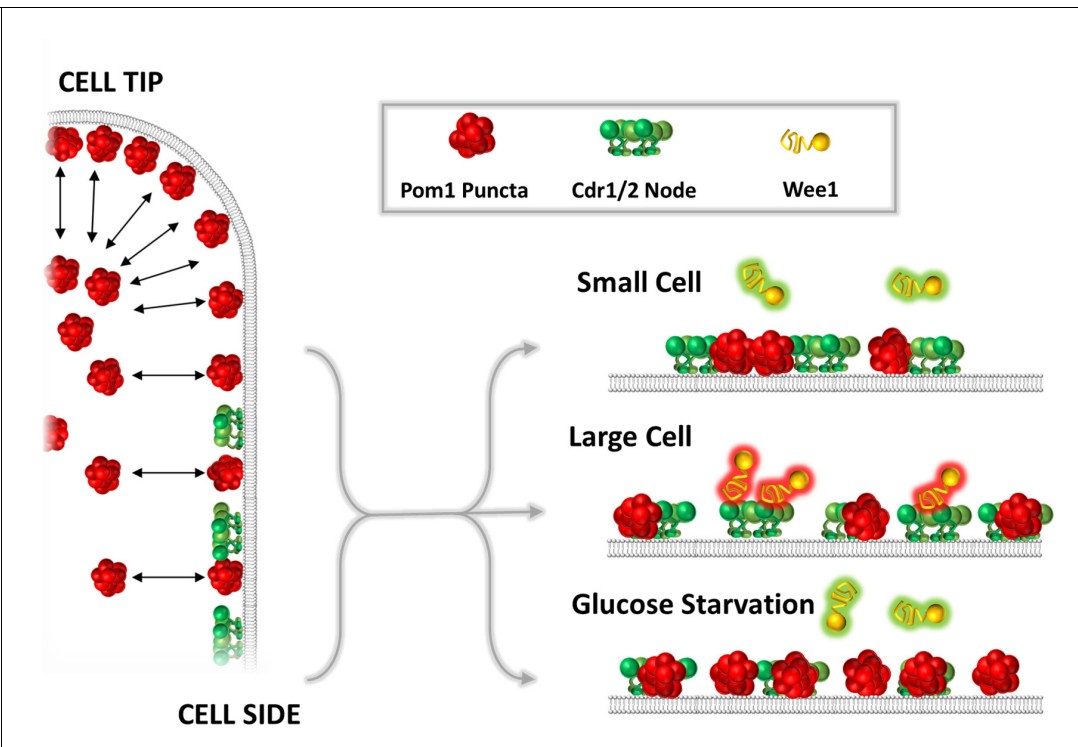

**Figure 8.** A model for the glucose-modulated control of Wee1 bursting by Pom1 and Cdr2. See text for discussion.
DOI: https://doi.org/10.7554/eLife.46003.022

and ensure that cells divide only when they have grown large enough. In this context, the Pom1 threshold is only strong enough to inhibit Cdr2 signaling in small cells, where node density is low. In this manner, Pom1 prevents precocious entry into mitosis. (2) Under glucose restriction, Pom1 and Cdr2 cooperate to integrate information about both cell size and glucose availability, and signal that information to the core cell cycle machinery.

### The Pom1 gradient as a read-out for cell polarity

More broadly, our study reveals the adjustable nature of the concentration gradient formed by Pom1 clusters. Concentration of Pom1 clusters at cell tips reads out the polarity state of the cell for downstream signaling by Cdr2 and Wee1. Conditions that alter the polarized growth state of the cell would lead to changes in the gradient of Pom1 clusters, resulting in altered cell size at division. This dynamic system appears to operate in a manner that depends on both cell size and nutrient availability. Identification of additional growth and environmental conditions that alter the relative distributions of Pom1 clusters and Cdr2 nodes, and determination of how they connect with other cell cycle signaling pathways, may reveal new mechanisms for nutrient modulation of cell size control.

### Integration of the Pom1-Cdr2-Wee1 pathway with other sizing mechanisms

Pom1-Cdr2-Wee1 signaling is cell size dependent but is not the only pathway that contributes to the overall cell size control network. Intriguingly, a recent study reported that $cdr2\Delta$ cells no longer divide based on accumulation of a threshold surface area and instead revert to a secondary mechanism of cell size control based on volume sensing (*Facchetti et al., 2019*). These results demonstrate that cells use different signal transduction networks to measure different aspects of their size. Whereas the Pom1-Cdr2 network is well suited to measure cortical surface area, size-control systems that measure other aspects of size would require different signaling logic. Theoretically, mechanisms for cell volume sensing could depend on proteins whose concentrations do not scale linearly with cell size. Such mechanisms are thought to underlie cell size control in other organisms, such as the budding yeast, where growth-dependent dilution of the transcriptional repressor Whi5 is thought to underlie $G_1/S$ size control (*Schmoller et al., 2015*). One possible candidate for cellular volume sensing in fission yeast is Cdc25, which increases in concentration with cell size (*Keifenheim et al., 2017*; *Moreno et al., 1990*). These mechanisms are attractive candidates for volume sensors because cytoplasmic concentration is independent of cell geometry. The existence of sizing mechanisms that measure other aspects of cell size, and function independently of the Pom1-Cdr2 surface area sensing network, underscores that cell size is a system level property controlled by multiple signaling pathways. Thus, it remains important to discover size-dependent signaling mechanisms within each individual pathway, as a step towards understanding the integration of multiple pathways into a larger size control system.

## Materials and methods

**Key resources table**

| Reagent type (species) or resource | Designation | Source or reference | Identifiers | Additional information |
|---|---|---|---|---|
| Strain, strain background (*S. pombe*) | *pom1-yomNeon Green::hphR* | This Paper | JM4496 | Integration through PCR product transformation |
| Strain, strain background (*S. pombe*) | *pom1-3HA::hphR ura4-D18 leu1-32 h+* | PN948 | JM797 | Paul Nurse Lab |
| Strain, strain background (*S. pombe*) | *pom1Δ::natR h-* | Lab Stock | JM966 | |

*Continued on next page*

*Continued*

| Reagent type (species) or resource | Designation | Source or reference | Identifiers | Additional information |
|---|---|---|---|---|
| Strain, strain background (*S. pombe*) | *pom1-m2-yomNeon Green::hphR ura4-D18 leu1-32 h-* | This Paper | JM5412 | Integration through PCR product transformation |
| Strain, strain background (*S. pombe*) | *pom1-3HA::hphR tea1Δ::ura4 + ura4-D18* | This Paper | JM5414 | Progeny from cross between JM797 and JM219 |
| Strain, strain background (*S. pombe*) | *pom1-3HA::hphR tea4Δ::kanMX6* | This Paper | JM5415 | Progeny from cross between JM797 and J M2256 |
| Strain, strain background (*S. pombe*) | *pom1-yomNeonGreen::hphR tea1Δ::kanMX6* | This Paper | JM4792 | Progeny from cross between JM4496 and JM838 |
| Strain, strain background (*S. pombe*) | *pom1-yomNeonGreen::hphR tea4Δ::kanMX6* | This Paper | JM4791 | Progeny from cross between JM4496 and JM218 |
| Strain, strain background (*S. pombe*) | *pom1-yomNeonGreen::hphR cdr2-tagRFP-t::hphR* | This Paper | JM5373 | Progeny from cross between JM4699 and JM4160 |
| Strain, strain background (*S. pombe*) | *cdr2-yomNeonGreen::hphR h-* | Lab Stock | JM4493 | |
| Strain, strain background (*S. pombe*) | *wee1-yomNeon Green::hphR cdr2Δ::natR* | Lab Stock | JM4525 | |
| Strain, strain background (*S. pombe*) | *cdr2-yomNeonGreen::hphR pom1-tdTomato::natR* | This Paper | JM5135 | Progeny from cross between JM935 and JM4493 |
| Strain, strain background (*S. pombe*) | *cdr2-mEGFP::kanMX6 ura4-D18 leu1-32 ade6-m210 h+* | Lab Stock | JM346 | |
| Strain, strain background (*S. pombe*) | *cdr2-yomNeon Green::hphR pom1Δ::natR* | This Paper | JM5359 | Progeny from cross between JM5238 and JM966 |
| Strain, strain background (*S. pombe*) | *wee1-yomNeon Green::hphR h-* | Lab Stock | JM4495 | |
| Strain, strain background (*S. pombe*) | *wee1-yomNeon Green::hphR pom1Δ::kanMX6* | Lab Stock | JM4527 | |
| Strain, strain background (*S. pombe*) | *972 h-* | PN1 | JM366 | Paul Nurse lab |
| Strain, strain background (*S. pombe*) | *cdr2Δ::natR ura4-D18 leu1-32 h+* | Lab Stock | JM600 | |
| Strain, strain background (*S. pombe*) | *wee1-yomNeon Green::hphR cdr2(E177A)* | Lab Stock | JM4578 | |
| Recombinant DNA reagent | pFA6a-yomNeon Green::hphR | Lab Stock | pJM1344 | |
| Recombinant DNA reagent | pFA6a-tdTomato::natR | Lab Stock | pJM248 | |
| Recombinant DNA reagent | pFA6a-yomTagRFP-T::hphR | Lab Stock | pJM1221 | Derived from Addgene Plasmid 44842 |
| Recombinant DNA reagent | pFA6a-mEGFP::kanMX6 | Lab Stock | pJM228 | |

*Continued on next page*

*Continued*

| Reagent type (species) or resource | Designation | Source or reference | Identifiers | Additional information |
|---|---|---|---|---|
| Recombinant DNA reagent | pFA6a-3HA::hphR | Lab Stock | pJM216 | |
| Antibody | anti-HA (mouse monoclonal) | Covance | MMS-101R | WB (1:3000) |
| Antibody | anti-GST (rabbit polyclonal) | Covance | custom | WB (1:3000) |
| Chemical compound, drug | 1,2-dioleoyl-sn-glycero-3-phospho-L-serine (sodium salt) | Avanti Polar Lipids | 840035C | 18:1 DOPS or 'PS' |
| Chemical compound, drug | 1,2-dioleoyl-sn-glycero-3-phosphoethanolamine-N-(lissamine rhodamine B sulfonyl) (ammonium salt) | Avanti Polar Lipids | 810150C | 18:1 Liss Rhod PE |
| Chemical compound, drug | 1,2-dioleoyl-sn-glycero-3-phosphocholine | Avanti Polar Lipids | 850375C | 18:1 (Δ9-Cis) PC (DOPC) or 'PC' |

## Strain construction and media

Standard *S. pombe* media and methods were used (*Moreno et al., 1991*). Strains used in this study are listed in the Key Resources Table. Gene tagging and deletion were performed using PCR and homologous recombination (*Bähler et al., 1998*). The mNeonGreen (mNG) sequence was used under license from Allele Biotechnology. Addgene Plasmid 44842 (pFA6a-link-yoTagRFP-T-SpHis5) was a gift from Wendell Lim and Kurt Thorn (Addgene plasmid # 44842; http://n2t.net/addgene:44842; RRID:Addgene_44842).

## TIRF microscopy and analysis

Pom1 clusters and node components were imaged using simultaneous dual-color total internal reflection fluorescence (TIRF) microscopy to limit excitation of fluorophores to those nearest to coverslip. Imaging was performed on a commercially available TIRF microscope (Micro Video Instruments) composed of a Nikon Eclipse Ti microscope base equipped with a 100x Nikon Apo TIRF NA 1.49 objective and a two-camera imaging adaptor (Tu-CAM, Andor Technlogy) containing a dichroic and polarization filters (Semrock FF580-FDi01−25 × 36, FF02-525/40-25, FF01-640/40-25) to split red and green signal between two aligned Andor iXon electron-multiplied CCD cameras (Andor Technology). Red/green beam alignment was performed prior to imaging using a TetraSpeck Fluorescent Microsphere size kit (Thermofisher).

Standard #1.5 glass coverslips were RCA cleaned before use to remove fluorescent debris. Cells were grown in EMM4S, and washed into fresh EMM4S immediately before imaging to remove autofluorescent debris resulting from overnight culture. Cells were imaged in EMM4S media on glass slides at ambient temperature. Individual slides were used for no more than five minutes to prevent cells from exhausting nutrients or oxygen. Agar pads were not used due to increased background fluorescence. Image analysis and processing was performed using ImageJ2 (NIH).

## 'Head-On' tip imaging

'Head-On' or 'Tip' imaging was performed using a protocol modified from *Dodgson et al. (2013)*. Custom micro-well coverslips or Ibidi Sticky-Slide VI channels (Ibidi, Martinsried, Germany) were coated with BS-I lectin (1 mg/mL in water) by incubation for 30 min and washed 3X with water to remove non-adherent lectin. Cells were then added to the imaging chambers and allowed to settle to the cover-slip bottom where they adhere to the lectin. Cells oriented with the long axis perpendicular to the coverslip were identified using brightfield prior to fluorescence imaging.

## Spinning disc microscopy and analysis

Spinning-disc confocal imaging was performed using a commercially available system (Micro Video Instruments, Avon, MA) featuring a Nikon Eclipse Ti base equipped with an Andor CSU-W1 two-camera spinning disc module, dual Zyla sCMOS cameras (Andor, South Windsor, CT) an Andor ILE

laser module, and a Nikon 100X Plan Apo ⅄ 1.45 oil immersion objective. Cells were imaged in EMM4S media on glass slides at ambient temperature unless otherwise noted.

## Airyscan Super-Resolution microscopy and analysis

To achieve maximum resolution and sensitivity, fluorescence intensity of Pom1 clusters on both cell sides and cell tips was measured using a Zeiss Airyscan microscope (*Figure 1—figure supplement 1C,D,E*), composed of a Zeiss LSM-880 laser scanning confocal microscope (Zeiss, Oberkochen, Germany) equipped with 100X alpha Plan-Apochromat/NA 1.46 Oil DIC M27 Elyra objective, Airyscan super-resolution module and GaAsP Detectors, and Zen Blue acquisition software using the Super-resolution mode with pin-hole size of 1.2 airy-units to prioritize resolution. Z-volumes of 16 slices with 0.19 µm spacing for high spatial resolution in all dimensions were centered on the cell cortex closest to the coverslip. Airyscan images were processed in Zeiss Zen Blue software, and quantification was performed on sum projections of Airyscan reconstructed stacks.

## Single particle tracking

Image analysis and processing was performed using ImageJ2 (NIH). Cdr2 node numbers and Pom1/Wee1 clusters number, frequency and binding duration were quantified using the Trackmate plugin (*Tinevez et al., 2017*) to analyze TIRF microscopy movies. Due to variable fluorescence intensity in different TIRF fields and images, thresholding parameters were determined separately for each image, and accuracy was confirmed by visual inspection to ensure that only nodes/clusters were counted and that no nodes/clusters were omitted. For Wee1 burst and Pom1 cluster tracking, Particle Diameter was set to 0.3 microns (approximate XY resolution), with Maximum Gap Linking set to two frames, and Linking Range for particle tracking was set to 0.15 microns. Lookup table was adjusted to fire for some images to emphasize signal intensities. For Pom1 cluster tracking, analysis was restricted to 2 µm x 2 µm square ROIs positioned at the cell middle or tip.

## Colocalization analysis

Colocalization analysis of Cdr2 nodes and Pom1 clusters as in *Figure 4F,G* and *Figure 4—figure supplement 2E,F* was performed using two-color simultaneous TIRF microscopy images as in *Figure 4A*. Analysis was restricted to 2 µm$^2$ square ROIs positioned at the cell middle. The spot detection algorithm in the Trackmate plugin (*Tinevez et al., 2017*) for ImageJ2 was used to assign 0.3 µm diameter circular ROIs to Cdr2 nodes (cyan) and Pom1 clusters (magenta) in images as in *Figure 4F*. Pom1 and Cdr2 structures were counted as colocalized if the centroids of their respective ROIs were spaced <150 nm apart (yellow ROI pairs in *Figure 4F*.

## Mean Squared Displacement

Mean Squared Displacement analysis and calculation of diffusion coefficients were performed using the MATLAB (MathWorks, Natick, MA) class @msdanalyzer developed in *Tarantino et al. (2014)*. Briefly, single-particle traces of high-speed (100 ms acquisition) TIRFM videos were generated using the Trackmate plugin for ImageJ2 as described above and were then exported to MATLAB for further analysis. @msdanalyzer was then used to plot and calculate MSD for each particle trace, plot the MSD curves, compute and fit the mean MSD curve, and compute the diffusion coefficient.

## Pom1 purification

The full length Pom1 sequence was purified as follows: Pom1 sequence was subcloned into pGEX6P1 vector using the Xho1 restriction site and expressed as a GST-fusion protein in *Escherichia coli* strain BL21(DE3). Transformants were cultured to log-phase at 37 ˚C, followed by a shift to 25 ˚C for 30 min. Expression was then induced by addition of 1-thio-β-D-galactopyranoside to 200 µM, followed by growth for an additional 3 hr at 25 ˚C. Cells were harvested by centrifugation and lysed by passing them twice through a French press in lysis buffer (1xPBS, 100 mM NaCl, 1 mM DTT, and complete EDTA-free protease inhibitor tablets (one tablet/50 mL Buffer) (Roche, Basel, Switzerland)). Following lysis, Triton-X 100 was added to 1% V/V. Lysates were then cleared by centrifugation for 10 min at 12,000 x g at 4 ˚C in a Sorval SS-34 fixed-angle rotor (Thermo Scientific, Waltham, MA). The supernatant was then incubated with glutathione-agarose (Sigma Aldrich, St. Louis, MO) for 2 hr

at 4 ˚C. 20 mM glutathione (pH 8.0) was used to elute purified protein, and the eluate was dialyzed overnight at 4 ˚C into 10 mM Tris-HCl, pH8 +5% V/V glycerol.

## Preparation of yeast extracts

Fission yeast detergent extracts were prepared by growing 1.5 L cells to mid-log phase (OD ~0.3), and then washed twice with 50 mL Node Isolation Buffer (NIB – 50 mM HEPES, 100 mM KCl, 1 mM MgCl$_2$, 1 mM EDTA, pH 7.5) (*Allard et al., 2018*). Next, we resuspended the pellet in an equal volume of 2X NIB (W/V) containing a protease/phosphatase inhibitor cocktail (10 µL/mL 200x PI, 50 µL/mL 1M β-glycerol phosphate, 50 µL/mL 1M NaF, 2.5 µL/mL 200 mM PMSF, 1 mM DTT), and snap froze the resuspension as pellets by pipetting drop-wise into a liquid nitrogen bath. Then, yeast pellets were ground using liquid-nitrogen chilled coffee-grinders for 2 min, and collected into chilled falcon tubes and stored at −80 ˚C. 1.5 g of frozen yeast powder was then thawed on ice, and Triton X-100 was added to a final concentration of 1%, and the extracts were mixed by gentle pipetting. Extracts were then centrifuged at 4 ˚C for 10 min at 20,000 x g to yield a low speed supernatant, which were then used for subsequent experiments.

## Sucrose gradient ultracentrifugation

Discontinuous sucrose gradients were prepared in 14 × 89 mm Ultra Clear Ultracentrifuge tubes (Beckman Coulter, Brea, CA) by layering 5–23% (top to bottom) sucrose in NIB + 1% Triton X-100 in 0.37 mL steps of 2% sucrose increments. 700 µL of yeast extract or recombinant protein diluted in NIB Buffer was then added to the top of the gradient. Sucrose gradients were centrifuged at 100 k x g (50 kRPM) in a Beckman L8-M ultracentrifuge for 2 hr at 4 ˚C in a chilled SW60ti swinging bucket rotor. 0.5 mL gradient fractions were collected from the top by hand, vortexed, and 100 µL of each fraction was mixed 2:1 in 3X SDS-PAGE sample buffer (65 mM Tris pH 6.8, 3% SDS, 10% glycerol, 10% 2-mercaptoethanol, 50 mM NaF, 50 mM β-glycerophosphate, 1 mM sodium orthovanadate) and boiled for 5 min. Samples were then subjected to SDS-PAGE and western blotting or Coomassie staining. To calculate S-values of sedimentation peaks from western blot signal intensities, mean band intensities were measured using Image Studio Lite (LICOR, Lincoln, NE), and Gaussian Curves were fit to these values. The peak of the Gaussian for each sedimentation peak of each protein was used to assign the known or interpolated S-value.

## Western blotting

For western blots, cells were lysed in 150 µl of 3X SDS-PAGE sample buffer with glass beads in a Mini-beadbeater-16 (Biospec, Bartlesville, OK) for 2 min. Gels were run at a constant 20 mAmps until 75 kDa marker was at the bottom of the gel. Blots were probed with anti-HA (Covance, Princeton, NJ) or homemade anti-GST. For monitoring Wee1 phosphorylation, samples were run on an SDS-PAGE gel containing 6% acrylamide and 0.02% bisacrylamide, and then probed with a homemade anti-Wee1 antibody (*Allard et al., 2018*). Rabbit anti-GST antibody (Covance) was raised against recombinant GST protein expressed and purified from *E. coli*.

## Supported Lipid Bilayers

To prepare supported lipid bilayers, we first prepared small unilamellar vesicles (SUVs) composed of three lipids: (1) DOPC (18:1 (Delta9) Cis PC - 1,2 Dioleoyl-sn-glycero-3-phosphocholine), (2) DOPS (18:1 PS - 1,2-Dioleoyl-sn-Glycero-3-[Phospho-L-Serine]), and (3) fluorescent 18:1 Liss Rhod PE (1,2-dioleoyl-sn-glycero-3-phosphoethanolamine-N-(lissamine rhodamine B sulfonyl). All lipids were dissolved in chloroform (Avanti, Alabaster, AL). 2µmol total lipid with the desired molar ratios were mixed in glass vials with PTFE coated caps, which had been RCA cleaned and rinsed three times with chloroform before use. Excess chloroform was evaporated in a vacuum desiccator for 1 hr. Lipids were resuspended in 400 µL SUV Buffer (50 mM HEPES, 100 mM KCl, 1 mM EDTA, 1 mM MgCl$_2$, pH7.5) for 5 mM lipid mixture stocks. The lipid mixtures were vortexed until cloudy, and then transferred to 1.5 mL Eppendorf tubes and subjected to ten freeze-thaw cycles using a liquid nitrogen bath and 32 ˚C hot plate, with 2 min of sonication in a sonicating water bath following each thaw cycle. Stocks were stored at −80 ˚C in 20 µL aliquots.

Supported lipid bilayers were made by adding 10 µL of SUVs to RCA cleaned custom microwells built on 22 × 40 mm #1.5 coverslips. Chambers were then incubated at 37˚C for one hour to induce

vesicle fusion, and unincorporated vesicles were rinsed away using five rinses with 100 μL SUV buffer. Yeast extract was added to these chambers by removing 90 μL of buffer, leaving just enough to cover the chamber bottom, and then adding extract.

## Statistics and line fitting

All plotting and statistical analysis was performed using GraphPad Prism 8.0.2 (263) (GraphPad Software, San Diego, CA), except for MSD analysis, which was performed using MATLAB as described above. Analysis of statistical significance was performed using unpaired Student's T-tests for comparison of two data sets where appropriate or using single-factor ANOVA with Tukey's multiple comparison test for comparison of >2 data sets. The specific test used for each experiment and the results of that test are listed in the figure legends, with one exception: Analysis of the data presented in *Figure 3C,D* is provided in an ANOVA table in *Figure 3—figure supplement 1*.

In general, line fitting was performed using linear regression by least-squares with analysis of residuals to ensure fit. Where linear regression did not fit the biological question, we compared of multiple models (linear regression vs non-linear regression, and the coefficient of determination ($R^2$) was considered to determine best-fit. When considering the regression model in *Figure 6C,D*, visualization of the non-log scale plots revealed unambiguous exponential relationships between glucose concentration and the concentration of either Pom1 or Cdr2, reflecting saturable binding. In *Figure 7B,C*, insets indicate whether the slopes ($b_1$) and Y-intercepts ($b_0$) of the two linear regressions are significantly different at the 95% confidence level. Details of the linear regressions presented in *Figure 7B,C* and *Figure 7—figure supplement 1A,B* are provided in *Supplementary file 1* and *Supplementary file 2*. To summarize the results of a given test, statistical significance is indicated within plots by the convention: (****) indicates $p < 0.0001$, (***) indicates $p < 0.001$, (**) indicates $p < 0.01$, (*) indicates $p < 0.05$, n.s. indicated $p > 0.05$.

## Acknowledgements

We thank members of the Moseley lab and Erik Griffin for comments on the manuscript. B Wickner, C Barlowe, the Life Sciences Center Imaging Facility, and the bioMT Molecular Interactions and Imaging Core at Dartmouth for sharing equipment. A Lavanway, Z Svindrych and the Life Sciences Light Microscopy Facility for assistance with microscopy. A Orr for assistance with ultracentrifugation. Y Wu for recombinant Pom1 proteins. This work was supported by grants from the American Cancer Society (RSG-15-140-01-CCG) and the NIH (R01 GM099774) to JBM; and an NIH Training Grant (T32GM008704) to CAHA. Shared imaging resources were supported by grants from the NIH (1S10OD018046 and P20GM113132).

## Additional information

### Funding

| Funder | Grant reference number | Author |
| --- | --- | --- |
| National Institute of General Medical Sciences | R01GM099774 | James B Moseley |
| American Cancer Society | RSG-15-140-01-CCG | James B Moseley |
| National Institute of General Medical Sciences | T32GM008704 | Corey A H Allard |

The funders had no role in study design, data collection and interpretation, or the decision to submit the work for publication.

### Author contributions

Corey A H Allard, Conceptualization, Formal analysis, Investigation, Methodology, Writing—original draft, Writing—review and editing; Hannah E Opalko, Investigation, Methodology, Writing—review and editing; James B Moseley, Conceptualization, Formal analysis, Supervision, Funding acquisition, Writing—original draft, Project administration, Writing—review and editing

Author ORCIDs
James B Moseley https://orcid.org/0000-0002-7354-7416

Decision letter and Author response
Decision letter https://doi.org/10.7554/eLife.46003.027
Author response https://doi.org/10.7554/eLife.46003.028

## Additional files

### Supplementary files

• Supplementary file 1. Parameters of linear regressions in *Figure 7B,C* and *Figure 7—figure supplement 1A,B*. For each data set, the table includes: The slope of the linear regression ($b_1$) and its 95% confidence intervals; whether the slope of the linear regression significantly deviates from zero at the 95% confidence interval; the Y-intercept of the linear regression ($b_0$) and its 95% confidence intervals; the coefficient of determination of the linear regression ($R^2$); the *n* of the data set.
DOI: https://doi.org/10.7554/eLife.46003.023

• Supplementary file 2. Summary of the statistical comparisons of the linear regressions in *Figure 7B, C* and *Figure 7—figure supplement 1A,B*. For each comparison, the table includes: Whether the slopes of the linear regressions ($b_1$) are significantly different from each other at the 95% confidence level; whether the Y-intercepts of the linear regressions ($b_0$) are significantly different from each other at the 95% confidence level. If the slopes of the linear regressions differed significantly, the difference in Y-intercepts was not tested.
DOI: https://doi.org/10.7554/eLife.46003.024

• Transparent reporting form
DOI: https://doi.org/10.7554/eLife.46003.025

### Data availability

All relevant data is included in the manuscript and supporting files.

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
