## [Decision Letter]

Thank you for submitting your article "Stable Pom1 clusters form a glucose-modulated concentration gradient that regulates mitotic entry" for consideration by *eLife*. Your article has been reviewed by three peer reviewers, and the evaluation has been overseen by a Reviewing Editor as the Mohan Balasubramanian and Naama Barkai as the Senior Editor. The reviewers have opted to remain anonymous.

The reviewers have discussed the reviews with one another and the Reviewing Editor has drafted this decision to help you prepare a revised submission.

The reviewers and the reviewing editor appreciated the area under investigation and were enthusiastic about your work on defining the mechanism of regulation of mitotic entry and cell size through the Pom1, Cdr2, and Wee1 network. In particular, your efforts towards biochemical reconstitution were seen as a strength of the manuscript.

However, a number of major issues have been raised which need to be addressed before this paper can be published in *eLife*. I hope you will be able to revise them to the satisfaction of the reviewers (although I hope I will be able to decide on my own in order to save time).

Essential revisions:

1) In agreement with previous reports, the presence of clusters at the cell tips and sides is dependent on Tea1 and Tea4. Interestingly Tea1 and Tea4 appear to regulate Pom1 clusters in slightly different ways. However, the individual contribution of *tea1* and *tea4* in regulating Pom1 clusters is not explored in this paper. A previous report has shown that upon glucose starvation Pom1 localization to the cell sides increases due to disruption of the microtubules and mislocalization of Tea4. In agreement with these previous reports, here the authors demonstrate that the frequency of Pom1 clusters at the cell sides increases in low glucose. The authors show that at the cell middle Pom1-Cdr2 nodes overlap suggesting that existence of such overlapping modes would lead to inhibition of cell division. Interestingly, in low glucose media Pom1 clusters at the cell middle increases while Cdr2 nodes here decrease. It is not clear how glucose regulates the number of *cdr2* nodes at the cell middle and how this contributes to cell size regulation in addition to the Pom1 gradient itself.

2) The model predicts that Pom1 clusters transiently visit cortical nodes on the plasma membrane to phosphorylate and inhibit Cdr2. One assumption of the model is that most of Pom1 and Cdr2 are on the plasma membrane, which were not mentioned or tested. If these data are available in the literature, they should be cited. If not, the authors can easily measure the fractions of proteins on the plasma membrane and cytoplasm using fluorescence intensity. Does cytoplasmic concentrations of Pom1 and Cdr2 change with cell size? If most of Pom1 and Cdr2 are in the cytoplasm, there is a possibility that Pom1 inhibits Cdr2 in the cytoplasm to regulate mitotic entry.

3) Pom1 is still in the stable cluster in the bleeding cells. Does Cdr2 form cluster in the cytoplasm? If yes, do Cdr2 and Pom1 colocalize in cytoplasmic clusters?

4) In several figures, TIRF images of cell cortex are compared to the middle focal plane of confocal images. The confocal images of cell cortex (near the plasma membrane) should be used for better comparison.

5) The diffusion of Pom1 puncta in Figure 1C and G can be plotted as mean square displacement (MSD) over time. If the MSD increases over time, it is consistent with 2D diffusion. The diffusion constant can be calculated to see if it is consistent with the 60S complex revealed by the sucrose gradient.

6) The paper definitely needs streamlining and improvements in writing. I think a good way of summarising this is to look at the figures. There are 9 main figures, all of which (bar the last one) contain significant quantities of data. This level of information, rather than helping, is actually hindering the paper. The authors have not focused clearly enough on what is the important data to show and how can it be shown in a concise and clear manner. This then feeds back into the text, where each of these figure panels then needs to be discussed. I don't want to prescribe precise changes, but the authors need to revisit the structure and try to build a story that is more precisely focused.

7) As with Gerganova et al., 2019. I have issues with the conclusion that Pom1 is regulating division at a precise length. The data strongly supports a role for *pom1* in stopping cells dividing before 11μm. However, I do not see clear evidence it is playing a role in promoting mitotic entry at a specific size (i.e. 14μm), despite what is written in the last line of Abstract.

8) Statistical analysis is weak at times. For example Figure 3—figure supplement 1 has error bars going into negative dwell times. This does not give confidence that the analysis was done rigorously or using the correct tools. Further, details of line fitting are vague and goodness of fit analysis is generally poor (e.g. fitting of sinusoidal curve to Supplementary Figure 6C and Figure 7.

Optional revision point:

1) A major strength of this paper is the authors ability to reconstitute Pom1 clusters in vitro. Ideally I would like to see the authors expand on this technique to determine how Pom1 clusters are formed. At the very least they can purify the clusters and use mass-spec to determine if post-translational modifications or interactions with other proteins leads to cluster formation.

---

## [Author Response]

Essential revisions:1) In agreement with previous reports, the presence of clusters at the cell tips and sides is dependent on Tea1 and Tea4. Interestingly Tea1 and Tea4 appear to regulate Pom1 clusters in slightly different ways. However, the individual contribution of tea1 and tea4 in regulating Pom1 clusters is not explored in this paper. A previous report has shown that upon glucose starvation Pom1 localization to the cell sides increases due to disruption of the microtubules and mislocalization of Tea4. In agreement with these previous reports, here the authors demonstrate that the frequency of Pom1 clusters at the cell sides increases in low glucose. The authors show that at the cell middle Pom1-Cdr2 nodes overlap suggesting that existence of such overlapping modes would lead to inhibition of cell division. Interestingly, in low glucose media Pom1 clusters at the cell middle increases while Cdr2 nodes here decrease. It is not clear how glucose regulates the number of cdr2 nodes at the cell middle and how this contributes to cell size regulation in addition to the Pom1 gradient itself.

In streamlining the manuscript (see essential revision #6 below), we have shortened our description of the *tea1*∆ and *tea4*∆ phenotypes. From our data, we can conclude that these polarity proteins provide the geometric patterning of Pom1 cluster binding to the cortex, resulting in a concentration gradient. We agree that additional exploration of their mechanistic contributions will provide more insight in the future.

We have rewritten the text to clarify how glucose regulates Pom1 and Cdr2. In low glucose, Pom1 clusters are no longer concentrated at cell tips, leading to higher levels in the cell middle. This redistribution does not impact the number of Cdr2 nodes, but reduces the number of Cdr2 molecules per node. We have revised the text to make this point more clearly: “Pom1 redistribution to the lateral cortex under low glucose is required to induce partial disassembly of each Cdr2 node…We conclude that the increasing frequency of Pom1 cluster binding at the lateral cell cortex leads to a reduced number of Cdr2 molecules per node.” These nodes are less active to regulate Wee1 compared to high glucose conditions, as seen by our measurements of Wee1 localization in Figure 7. These results provide a mechanistic explanation for the previously observed role of Pom1-Cdr2 in controlling cell size under low glucose conditions (Kelkar and Martin, 2015).

2) The model predicts that Pom1 clusters transiently visit cortical nodes on the plasma membrane to phosphorylate and inhibit Cdr2. One assumption of the model is that most of Pom1 and Cdr2 are on the plasma membrane, which were not mentioned or tested. If these data are available in the literature, they should be cited. If not, the authors can easily measure the fractions of proteins on the plasma membrane and cytoplasm using fluorescence intensity. Does cytoplasmic concentrations of Pom1 and Cdr2 change with cell size? If most of Pom1 and Cdr2 are in the cytoplasm, there is a possibility that Pom1 inhibits Cdr2 in the cytoplasm to regulate mitotic entry.

We performed the suggested experiments, and the results are presented in the new Figure 4—figure supplement 1. For both Pom1 and Cdr2, we measured the total cortical and cytoplasmic levels, and also examined the cortical levels at cell tips versus medial cortex. For Cdr2, our results are consistent with past work published in Pan et al., 2014. In short, both Pom1 and Cdr2 are strongly enriched at the cortex versus cytoplasm, and the cytoplasmic concentration of each protein is unaffected by cell size. In addition, we did not detect colocalization of Cdr2 nodes and Pom1 clusters in cell extracts lacking membrane bilayers (new Figure 2—figure supplement 1D). These combined data strongly implicate the cortex as the site of signal transduction between Pom1 and Cdr2.

3) Pom1 is still in the stable cluster in the bleeding cells. Does Cdr2 form cluster in the cytoplasm? If yes, do Cdr2 and Pom1 colocalize in cytoplasmic clusters?

We performed the suggested experiment, and the results are shown in new Figure 2—figure supplement 1D. We have previously shown that Cdr2 nodes are stable in cell extracts (Allard et al., 2018). Based on our model, we would expect that Pom1 puncta and Cdr2 nodes would not stably associate in extruded cytoplasm. Following the reviewer’s suggestion, we made extracts from cells co-expressing Pom1-mNeonGreen and Cdr2-tagRFP-t. In TIRF microscopy experiments, we found that Cdr2 nodes and Pom1 clusters do not colocalize in these cytoplasmic extracts, supporting our conclusion that this interaction occurs at the cell cortex.

4) In several figures, TIRF images of cell cortex are compared to the middle focal plane of confocal images. The confocal images of cell cortex (near the plasma membrane) should be used for better comparison.

We have included confocal images at the cell cortex for comparison to TIRF, as suggested. The key figure with this comparison is Figure 1—figure supplement 1, which shows Pom1 clusters by confocal imaging. Several other confocal microscopy images are included in the manuscript, but are not used for direct comparison to the TIRF images. In these cases (e.g. Figure 3—figure supplement 1, and Figure 5—figure supplement 1), we have included middle focal planes because they show the most relevant view for the given experiments.

5) The diffusion of Pom1 puncta in Figure 1C and G can be plotted as mean square displacement (MSD) over time. If the MSD increases over time, it is consistent with 2D diffusion. The diffusion constant can be calculated to see if it is consistent with the 60S complex revealed by the sucrose gradient.

We thank the reviewers for this helpful insight, which has been added into the revised manuscript. We used the TIRFM particle traces from figure 1 to calculate MSD (see new Figure 1—figure supplement 2). Whereas a linear relationship between mean MSD and time would indicate free diffusion, we obtained a correlation that was initially linear but saturated at time >2 seconds, indicated constrained diffusion. Thus, Pom1 clusters appear to be restricted to diffusion within sub-micron “corrals”.

We extended this approach to purified Pom1-mNG clusters diffusing on artificial supported lipid bilayers. MSD measurements revealed that Pom1 clusters exhibit the same diffusion coefficient in vitro and in cells. However, in vitro diffusion on SLBs is no longer restricted within corrals, suggesting that some cellular feature restricts diffusion of Pom1 clusters. We have included two new figures containing these data: Figure 1—figure supplement 2 (in vivo) and Figure 2—figure supplement 3 (in vitro).

6) The paper definitely needs streamlining and improvements in writing. I think a good way of summarising this is to look at the figures. There are 9 main figures, all of which (bar the last one) contain significant quantities of data. This level of information, rather than helping, is actually hindering the paper. The authors have not focused clearly enough on what is the important data to show and how can it be shown in a concise and clear manner. This then feeds back into the text, where each of these figure panels then needs to be discussed. I don't want to prescribe precise changes, but the authors need to revisit the structure and try to build a story that is more precisely focused.

We appreciate this constructive criticism, and have made an effort to shorten the manuscript without sacrificing clarity. Based on this comment, we have removed the data and discussion for Pom1 clusters and activity in *tea1∆* cells (Figure 5 and supporting supplementary data in the previous submission), which we realize were tangential to the main conclusions of the paper. We have also removed some data and discussion that were replications of past work (e.g. Figure 1—figure supplement 1 from the past submission). We have also tried to streamline the text to maintain a constant narrative.

7) As with Gerganova et al., 2019 I have issues with the conclusion that Pom1 is regulating division at a precise length. The data strongly supports a role for pom1 in stopping cells dividing before 11μm. However, I do not see clear evidence it is playing a role in promoting mitotic entry at a specific size (i.e. 14μm), despite what is written in the last line of Abstract.

We agree that this is an important point. In our initial submission, the Abstract stated that the Pom1-Cdr2-Wee1 pathway promotes mitotic entry at a specific size. We did not intend to imply that Pom1 performs a function specifically when cells reach that size. Rather, we would like to state that the entire pathway helps to specify the size at which cells divide, as cell size is altered in mutants of this pathway. In an attempt to clarify this point, we have revised the Abstract to read that “Pom1-Cdr2-Wee1 operates in multiprotein clusters at the cortex to promote mitotic entry at a cell size that can be modified by nutrient availability.”

8) Statistical analysis is weak at times. For example Figure 3—figure supplement 1 has error bars going into negative dwell times. This does not give confidence that the analysis was done rigorously or using the correct tools. Further, details of line fitting are vague and goodness of fit analysis is generally poor (e.g. fitting of sinusoidal curve to Supplementary Figure 6C and Figure 7.

We have made an effort to explain and to justify our statistical analyses with additional detail and text in the Materials and methods section. This added text also explains methods and rationale for curve fitting. We have also added the new Figure 3—figure supplement 1D, which reports all of the statistical data for Pom1 cluster dynamics in *tea1*∆ and *tea4*∆ mutants, and Supplementary files 1 and 2, which report the parameters of the linear regressions in Figure 7 and Figure 7—figure supplement 1, and a summary of the statistical comparisons in those figures. Many of the parameters that we have measured exhibit clear variability and biological noise, which we would argue should not be misconstrued as lack of experimental rigor. For example, in Figure 3—figure supplement 1B, most puncta bind briefly (200-400msec) but a subset bind for multiple seconds (i.e. orders of magnitude above the statistical mode). We did not wish to ignore the rare, longer binding events as “outliers,” and therefore have included all the data in our analyses. With these longer binding events included, the error bars should extend into negative dwell times. We did not mean to imply that negative dwell times should be considered as part of the underlying biological system, and therefore have edited this figure to show error bars extending in only one direction. We also understand that the coefficient of determination (R^2^) is low in many cases. However, these values do not reflect poor analysis, but rather we are specifically trying to highlight the *absence* of measurable size-dependence in a trend (slope not significantly different from 0, e.g. Figure 4—figure supplement 1D, E). By definition, this means that a linear model does not significantly improve on simply reporting an average, and thus yields a low R^2^. In these cases, a low R^2^ does not indicate low data quality, but rather represents the underlying biology.

Optional revision point:1) A major strength of this paper is the authors ability to reconstitute Pom1 clusters in vitro. Ideally I would like to see the authors expand on this technique to determine how Pom1 clusters are formed. At the very least they can purify the clusters and use mass-spec to determine if post-translational modifications or interactions with other proteins leads to cluster formation.

We have added new data (new Figure 2—figure supplement 3) that describe the diffusion of purified Pom1 clusters on supported lipid bilayers. However, we feel that additional characterization of Pom1 clusters (e.g. mass spec to identify new components and post-translational modifications) represents a key goal for the future, and would detract from the message of this paper. In this revision, we have attempted to streamline the paper by removing text and data that seem tangential to the central message, in response to essential revision #6. We hope to learn more about the composition, lipid binding preference, and biophysical properties of Pom1 clusters going forward. However, we believe that experiments to test each of these topics would open new avenues that would only make the paper more complicated at this stage. We share the enthusiasm of the reviewers and editors for this topic, and look forward to opening new lines of research with these analyses in the future.